# *Mandragora autumnalis*: Phytochemical Composition, Antioxidant and Anti-Cancerous Bioactivities on Triple-Negative Breast Cancer Cells

**DOI:** 10.3390/ijms26178506

**Published:** 2025-09-01

**Authors:** Ghosoon Albahri, Adnan Badran, Heba Hellany, Serine Baydoun, Rola Abdallah, Mohamad Alame, Akram Hijazi, Marc Maresca, Elias Baydoun

**Affiliations:** 1Doctoral School of Science and Technology-Platform of Research and Analysis in Environmental Sciences (EDST-PRASE), Beirut 1107, Lebanon; ghosoon.albahri.1@ul.edu.lb (G.A.); alamefs@hotmail.com (M.A.); akram.hijazi@ul.edu.lb (A.H.); 2Department of Biology, Faculty of Arts and Sciences, American University of Beirut, Riad El Solh, Beirut 1107, Lebanon; he115@aub.edu.lb (H.H.); rha62@mail.aub.edu (R.A.); 3Department of Nutrition, University of Petra Amman Jordan, Amman P.O. Box 961343, Jordan; abadran@uop.edu.jo; 4Breast Imaging Section, Imaging Institute, Cleveland Clinic Foundation, Cleveland, OH 44195, USA; baydous@ccf.org; 5Aix Marseille Univ, CNRS, Centrale Marseille, iSm2, 13013 Marseille, France

**Keywords:** *Mandragora autumnalis*, phytochemistry, antioxidant, anti-cancerous activity, triple negative breast cancer cells

## Abstract

Breast cancer is a common and chronic condition, and despite improvements in diagnosis, treatment, and prevention, the number of cases of breast cancer is rising annually. New therapeutic drugs that target specific checkpoints should be created to fight breast cancer. *Mandragora autumnalis* possesses substantial cultural value as a herb and is regarded as one of the most significant medicinal plants; however, little is known about its anticancerous biological activity and chemopreventive molecular pathways against the triple-negative breast cancer (MDA-MB-231) cell line. In this study, the antioxidant, anticancer, and underlying molecular mechanisms of the *Mandragora autumnalis* ethanolic leaves extract (MAE) were evaluated, and its phytochemical composition was determined. Results indicated that MAE diminished the viability of MDA-MB-231 cells in a concentration- and time-dependent manner. Although MAE exhibited 55% radical scavenging activity at higher concentrations in the 2,2-diphenyl-1-picrylhydrazyl (DPPH) assay, the attenuation of its cytotoxic effects in MDA-MB-231 cells with N-acetylcysteine (NAC) co-treatment suggests a potential role of oxidative stress. Additionally, MAE caused an increase in the tumor suppressor p53. Moreover, this extract caused a significant decrease in the expression of Ki-67 (a cellular proliferation marker), MMP-9 (matrix metalloproteinase-9, an enzyme involved in extracellular matrix degradation and metastasis), and STAT-3 (a transcription factor regulating cell growth and survival). Also, MAE altered cell cycle, cell migration, angiogenesis, invasion, aggregation, and adhesion to suppress cellular processes linked to metastasis. All of our research points to MAE’s potential to function as an anticancer agent and opens up new possibilities for the development of innovative triple-negative breast cancer treatments.

## 1. Introduction

Breast cancer is a prevalent malignant tumor that affects women [1]. A multitude of internal and external factors influence the development and incidence of breast cancer [2]. The milk duct is often the site of emergence; lobules are the site of less serious cases. Cancers that affect the ductal region are referred to as ductal carcinomas, and those that affect the mammary lobules are known as lobular carcinomas [3]. Triple-negative breast cancers (TNBCs) are classified as aggressive variants of breast cancer, characterized by diminished expression of the human growth factor receptor 2 and the progesterone and estrogen receptors [4]. TNBCs are responsible for 12–17% of all cases of breast cancer and naturally recur. They exhibit relatively more aggressive clinical behavior in comparison to other subtypes of breast cancer. Furthermore, 24% of newly diagnosed breast cancers are TNBCs, and their incidence has been steadily increasing. These cancers also have poor prognoses and typical metastatic patterns, where the average survival rate is approximately 10.2 months; for regional tumors, the 5-year survival rate is 65%, and for tumors that have spread to distant organs, it is 11% [5,6,7]. Globally, the incidence of breast cancer is increasing along with the disease’s burden, making breast cancer an important public health concern [8]. The incidence of breast cancer is expected to rise by over 40% by 2040, to approximately 3 million cases per year, as a result of population growth and aging. In a similar vein, over 50% more people will die from breast cancer in 2040 [9]. Several known risk factors, including late menopause, advanced age at first birth, fewer children, reduced breastfeeding, menopausal hormone replacement therapy, oral contraceptives, alcohol use, and excess body weight, have been associated with breast cancer [10].

The MDA-MB-231 cell line was chosen for this study due to its high clinical relevance as a representative model of TNBC. TNBC constitutes 15–20% of all breast cancer cases and is associated with rapid progression, early metastasis, poor prognosis, and the absence of effective targeted therapies [11]. MDA-MB-231 cells exhibit a basal-like molecular profile and harbor *TP53* mutations, closely reflecting the genetic landscape of TNBC tumors in patients [12]. Functionally, they possess a highly invasive and metastatic phenotype, characterized by mesenchymal morphology, elevated expression of pro-metastatic mediators such as MMP-9, and activation of oncogenic signaling pathways, including STAT3 [13]. Additionally, their well-documented chemoresistance to conventional therapies mirrors the therapeutic challenges faced in clinical TNBC management, making them a stringent and predictive platform for evaluating novel anticancer agents [14]. These combined genetic, phenotypic, and clinical parallels underscore the suitability of MDA-MB-231 cells as an optimal in vitro model for mechanistic investigations and preclinical testing of potential TNBC-targeted treatments.

On the other hand, the use of phytonutrients, also known as nutraceuticals and herbal medicines, is still growing worldwide, with many people turning to these products in various national healthcare settings to treat different health issues. For years, herbal remedies have been used as anticancer medications. Numerous anticancer ingredients including anti-inflammatory properties have been demonstrated to have both direct and indirect cytotoxic effects on the tumor microenvironment, cancer immunity, and the progression of chemotherapy [15,16]. From 200 to 1800 AD, according to Aristotle and Galen’s teachings, cancer was thought to be caused by the coagulation of “black bile.” Until recently, when the prevalence of biology has led to a 25% decrease in mortality, herbs were crucial in managing cancer symptoms and improving the quality of life and survival of patients with the disease [17]. Compared to conventional chemical drugs, herbal medicine fights cancer in a significantly different way, as it prevents DNA mutation in surviving cells. Another benefit of herbal medicines is that they can create an environment that is unfavorable to the growth of cancer [18].

The *Mandragora autumnalis* plant has been prized as one of the most significant medicinal plants and has had great cultural significance as an herb since ancient times. Its use throughout history has been remarkable. *Mandragora* has a wide range of uses, including medicinal, hallucinogenic, and boosting ovulation. Furthermore, *Mandragora autumnalis* possesses a narcotic effect, which is most likely caused by the presence of an alternative form of alkaloids [19]. The plant genus *Mandragora*, a member of the Solanaceae family, is commonly known as “mandrake”. *Mandragora* species has a long history of use as a traditional herbal remedy, as evidenced by the use of its roots, fruits, and leaves to treat conditions like ulcers, inflammation, insomnia, and eye disorders [20,21,22]. The plant’s biological activities and phytochemical composition, especially its anticancer potential, have not been thoroughly studied despite its historical significance. In order to address this gap, the current study characterized the phytochemical profile of *Mandragora autumnalis* ethanolic leaf extract (MAE) using qualitative assays and Liquid Chromatography-Mass Spectrometry (LC-MS). MAE’s anticancer activity against the MDA-MB-231 cell line was assessed, building on the plant’s established bioactive qualities and traditional therapeutic relevance. Also, the effect of MAE on cell proliferation, apoptosis induction, and several molecular markers of metastasis, such as adhesion, invasion, cell cycle regulation, angiogenesis, migration, and aggregation, was tested. The historical medicinal use of the plant, its phytochemical components, and its possible mechanisms for preventing the progression of TNBC are all directly linked by this integrative approach.

## 2. Results

### 2.1. Phytochemical Composition and Total Phenolic and Total Flavonoid Content (TPC and TFC) Assays

The presence of various primary and secondary bioactive metabolites, including terpenoids, tannins, fatty acids, flavonoids, phenols, and steroids, was revealed by a qualitative phytochemical screening of the MAE crude extract (Table 1).

A large class of secondary metabolites known as polyphenolic compounds is responsible for most plants’ antioxidant qualities. Our results showed that the TPC of MAE is 58.98 ± 7.40 mg (gallic acid equivalents) GAE/g and TFC is 36.47 ± 0.87 mg (quercetin equivalents) QE/g, as shown in Table 2.

### 2.2. LC-MS Analysis of MAE

LC-MS analysis of MAE in both positive and negative ionization allowed the determination of the retention times (RT) and mass-to-charge ratios (*m*/*z*) of 34 compounds. The LC-MS analysis in positive ionization mode (Table 3A, Figure 1) revealed a chemically diverse profile consisting of multiple classes of bioactive compounds. The most abundant group was alkaloids, represented by hyoscyamine, tropine, methylisopelletierine, tropinone, and solacaproine. The phenolic compounds group was marked by the presence of chlorogenic acid and flavonoids, which were identified through the detection of quercetin and chrysin. Fatty acid derivatives were another major group, including hexadecanamide (palmitic amide), ethyl palmitate, ethyl oleate, linoleic acid, and ethyl linolenate. Esters such as ethyl 3-hydroxybutanoate, 3-(methylthio)propyl acetate, and ethyl hydrocinnamate were also detected in notable quantities. Additionally, furan derivatives like 5-hydroxymethyl-2-furancarboxaldehyde and ketones and aldehydes such as 1-hydroxy-2-propanone, acetate, and β-ionone were present. A unique compound, simulanoquinoline, representing the quinoline alkaloids, was also observed. On the other hand, among the principal compounds detected under negative ionization mode (Table 3B, Figure 1) were polyphenolic acids and their derivatives, including chlorogenic acid with a high intensity, and caffeic acid. Also, Linolenic acid, a polyunsaturated fatty acid, coumarins with their derivatives, including scopoletin, and 4-Methylumbelliferyl acetate were present. Furthermore, flavonoids and glycosides, including rutin and hyperoside, were identified. All together, these results show MAE has a rich phytochemical profile, bolstering its potential as a source of bioactive compounds for additional pharmacological research.

### 2.3. MAE Has a Potent Radical Scavenging Potential

The results of this investigation demonstrated concentration-dependent free-radical scavenging activity for MAE (Figure 2). MAE demonstrated a high reducing power, where the free radical scavenging activity of MAE at a concentration of 600 µg/mL was 54.94 ± 0.89%. Qualitative analyses of the MAE revealed substantial levels of phenolic and flavonoid compounds, as confirmed by TPC, TFC, and LC-MS data. These findings suggest that higher concentrations of these phytochemicals may be linked to enhanced antioxidant activity in the DPPH assay.

### 2.4. MDA-MB-231 Cell Proliferation Is Markedly Inhibited by MAE

To examine the potential anti-proliferative impact of MAE on breast cancer, different concentrations of each extract (0, 50, 100, 200, 400, and 600 μg/mL) were examined at 24, 48, and 72 h of treatment. Results show a concentration- and time-dependent decrease in cell viability following MAE treatment. After 48 hr of treatment with 50, 100, 200, 400, and 600 µg/mL of MAE, the proliferation of MDA-MB-231 cells was 92.18 ± 7.32, 66.66 ± 8.82, 22.67 ± 0.17, 17.19 ± 0.92,18.00 ± 1.42%, respectively (Figure 3A). Using the 3-(4,5-dimethylthiazol-2-yl)-2,5-diphenyltetrazolium bromide (MTT) assay, the inhibitory concentration 50% IC_50_ of MAE was 467.28, 132.33, and 109.4 µg/mL at 24, 48, and 72 h. Based on that and the IC_50_ values, 100 and 200 µg/mL of MAE were used in further experiments. Crucially, the viability assay against MDA-MB-231 cells revealed that doxorubicin (DOXO) was more harmful to MDA-MB-231 cells with a (cytotoxic concentration 50%) CC_50_ of 0.0036 µg/mL after 72 h (Figure 3B).

To examine the potential correlation between the antiproliferative effects of the extracts and the generation of reactive oxygen species (ROS) induced by MAE, MDA-MB-231 cells were pretreated with N-acetyl cysteine (NAC), a ROS scavenger [23]. The findings indicated that NAC significantly reduced NAC-mediated cell death (Figure 3C). Specifically, the viability of cells treated with 100 μg/mL MAE for 24 h was 64.33% ± 0.007% in the absence of NAC and was significantly increased to 165.36% ± 0.001% when cells were pre-treated with NAC. Similarly, cells treated with 200 μg/mL MAE had a viability of 36.25% ± 0.002% without NAC, and cell viability was significantly increased to 281.47% ± 0.01% with NAC pretreatment. These findings suggest that MAE inhibits TNBC cell proliferation via a ROS-dependent mechanism.

Protein lysates from MAE-treated cells were immunoblotted with an anti-Ki67 antibody to confirm the anti-proliferative qualities of MAE, as this protein is associated with the growth of cancer cells. Ki67 is known to be an indicator and diagnostic marker for numerous cancers, and it is highly expressed in TNBC [24]. The treatment of MDA-MB-231 cells with MAE at 200 µg/mL significantly reduced Ki67 protein levels in a concentration-dependent manner 0.47 ± 0.07 fold, in comparison to the vehicle-treated control cells, as shown in Figure 3D. These findings demonstrate that MAE inhibits MDA-MB-231 cells’ ability to proliferate by interfering with their growth.

### 2.5. MAE Triggers Intrinsic Apoptosis in MDA-MB-231 Cells

Following a 24 h MAE treatment, MDA-MB-231 cells were examined using an inverted phase-contrast microscope. The total number of cells per microscopic field decreased in a concentration-dependent manner, according to the morphological observations of the MAE-treated cells. Furthermore, cells treated with MAE showed signs of apoptosis. Higher magnifications revealed apoptotic bodies, cytoplasmic shrinkage, membrane blebbing, loss of epithelial morphology, and rounded cell shapes (Figure 4A). Additional examination of MAE-treated cells using fluorescent stain 4′,6-diamidino-2-phenylindole (DAPI) staining demonstrated the development of apoptotic bodies, chromatin lysis, and nuclear condensation (Figure 4B). Next, we aimed to confirm that cells treated with MAE undergo activated apoptosis. To achieve this, first, we measured Procaspase-3 levels, which are crucial for apoptosis [25]. Western blotting was used to analyze the protein levels of the apoptosis effector enzyme Procaspase-3 and its cleavage products in order to confirm that MAE activated apoptosis. The results showed reduced levels of the Procaspase-3 protein in a concentration-dependent manner.

The levels of Procaspase-3 in cells treated with MAE at 200 μg/mL decreased significantly, reaching 0.54 ± 0.12-fold (Figure 4C). The levels of cleaved Caspase 3 fragments were also markedly elevated by MAE treatment concurrently. There was a notable rise in Caspase 3 cleavage products, with a 1.27 ± 0.02 increase at 200 μg/mL of MAE. This implies that the intrinsic cascade of apoptosis was initiated by MAE extracts, which caused the proteolytic cleavage of Procaspase-3 into its active form, Caspase 3.

It is known that B-cell lymphoma 2 (Bcl-2) is another anti-apoptotic protein that is crucial for the intrinsic apoptosis pathway [26]. According to our findings, MAE significantly reduced Bcl-2 protein levels to reach 0.70 ± 0.05 decrease at 200 µg/mL of MAE compared to the control (Figure 4C). Another apoptosis regulator is the pro-apoptotic protein Bcl-2-associated X-protein (Bax) [27]. In our investigation, cells treated with 200 µg/mL MAE showed significantly higher Bax levels with a 1.64 ± 0.02 fold increase (Figure 4C). These findings suggest that MAE targets apoptotic pathways to mediate its anticancer effects.

### 2.6. MAE Enhances the Aggregation of MDA-MB-231 Cells

One of the characteristics of cancer progression towards metastasis is the epithelial–mesenchymal transition (EMT), a process in which epithelial cells take on a mesenchymal phenotype marked by increased migratory and invasive capabilities and decreased cell–cell adhesion [28]. An aggregation assay was used to assess how MAE affected the MDA-MB-231 cells’ cell–cell adhesion characteristics. As demonstrated in Figure 5, there was a concentration-dependent rise in cell–cell aggregates following the treatment of cells with 100 and 200 µg/mL MAE, with significant increases of 73.39 ± 12.35 and 80.80 ± 8.86%, respectively. E-cadherin, also referred to as epithelial cadherin, is a key component of adherens junctions, which are specialized structures that promote strong cell–cell adhesion in epithelial tissues [29]. The downregulation of E-cadherin, specifically in TNBC, promotes the migration of breast cancer cells through the extracellular matrix, their invasion of blood and lymphatic vessels, and the disease’s dissemination to other organs. E-cadherin protein levels were raised in MDA-MB-231 cells treated with 100 and 200 µg/mL MAE in a dose-dependent manner, and at 200 µg/mL, there was a significant discernible difference from the control, with a 1.67 ± 0.22-fold change.

### 2.7. MAE Lowers the Adhesion Potential of MDA Cells

Integrins, among other members of the adhesion molecule family, are the primary mediators of cell-ECM(extracellular matrix) interaction. Elevated integrin β1 expression confers survival benefits to cancer cells and is linked to their capacity for migration and metastasis [30]. As seen in Figure 6A, the treatment of cells with 100 and 200 µg/mL MAE significantly reduced their adhesive abilities to 53.14103 ± 0.08237 and 48.45218 ± 0.031702%, respectively, compared to the control cells. Here, the application of 100 and 200 µg/mL MAE to MDA-MB-231 cells resulted in a significant decrease in the concentration-dependent increase in integrin β1 protein levels by 0.68 and 0.60-fold, contrasting in turn with the control cells that were subjected to the vehicle (Figure 6B). All of these findings support the hypothesis that MAE disrupts the integrin-ECM axis, which may limit MDA-MB-231 cells’ ability to spread.

### 2.8. MAE Decreases MDA Cells’ Ability to Migrate

Cell migration is crucial during physiological processes like immune responses, wound healing, and embryonic morphogenesis. It is also an important process that is a key characteristic of a malignant phenotype and plays a significant role in the early stages of cancer metastasis [31]. Using the wound-healing assay, the impact of MAE on MDA-MB-231 cell migration was investigated. MAE reduced the MDA-MB-231 cells’ ability to migrate and fill the wound in a concentration- and time-dependent manner, as shown in Figure 7A. For instance, treating MDA-MB-231 cells with 100 and 200 μg/mL of MAE significantly decreased their migration ability by 0.59 ± 0.13 and 0.40 ± 0.07 fold, respectively, compared to the control cells. These results were further confirmed by the transwell migration assay. MAE treatment significantly reduced MDA-MB-231 cells’ capacity to migrate from the upper to the lower chamber Figure 7B.

### 2.9. MAE Inhibits the Invasive Properties of MDA Cells

The invasion of other tissues is a crucial stage in the early stages of the cancer metastatic cascade, when cancer cells move from the primary tumor site and infiltrate secondary sites [32]. When compared to the control, MAE treatment significantly lowered the invasive potential of MDA-MB-231 cells in a dose-dependent manner using the Matrigel-coated trans-well chambers by 0.45 ± 0.13 and 0.26 ± 0.03 fold for 100 and 200 µg/mL, respectively (Figure 8).

STAT3 is a well-known and important participant in the development of tumors. Thus, focusing on the STAT3 pathway is a viable approach for creating new cancer medications because it stimulates the migration and invasion of cancer cells, prevents apoptosis, and promotes cell proliferation [33]. We measured the amount of the active phosphorylated form of STAT3 in MAE-treated MDA-MB-231 cells to see if *Mandragora autumnalis* mediates its anticancer activity by targeting the STAT3 pathway. Our findings demonstrated that, in comparison to the control, MAE treatment at 100 and 200 µg/mL significantly decreased the levels of STAT3 by 0.65 ± 0.08 and 0.48 ± 0.11 fold, respectively (Figure 8). This finding implies that the downregulation of the STAT3 signaling pathway is a necessary component of the MAE-mediated impact of TNBC progression and metastasis. Also, one widely known mechanism that encourages the migration and invasion of cancer cells is the breakdown of the ECM by matrix metalloproteinases (MMPs) [34]. MMP-9 activity levels were significantly reduced by 200 μg/mL MAE with a 0.57 fold change in comparison to vehicle-treated cells. These findings suggest that MAE may inhibit MMPs to lessen the invasive potential of MDA-MB-231 cells.

### 2.10. MAE Causes MDA-MB-231 Cells to Arrest During the Cell Cycle’s G_0_/G_1_ Phase

After treating MDA-MB-231 cells with 200 µg/mL of MAE for 24 h, flow cytometry was used to examine the cell cycle distribution. As seen in Figure 9A, the percentage of cells in the G_0_ phase rose in MAE-treated cells (12.3 ± 2.3 versus 5.3 ± 1.7 in control cells). The percentage of cells in the G_1_ phase increased with this (53.2 ± 5.8 vs. 29 ± 7.9 in control cells). The findings imply that MAE promotes cell cycle arrest during the G_0_/G_1_ phase. The percentage of cells in the S and G_2_-M phases was lowered along with this by 15.7 ± 2.7 and 17.6 ± 3.4 for control cells and 3.4 ± 1.2 and 8.2 ± 3.14, respectively, indicating that MAE causes cell cycle arrest at the G_1_ phase and inhibits cell entry into the S phase.

The p53 gene is a tumor suppressor; that is, when it is active, tumors cannot grow [35]. Phosphorylated p53 levels rose in response to MAE treatment in a concentration-dependent manner; at 200 µg/mL, there was a significant increase (1.53 ± 0.02-fold change in the control) (Figure 9B). This data implies that cell cycle arrest may result from MAE-induced p53 activation. Furthermore, the p38 MAPK (mitogen-activated protein kinase) pathway is commonly linked to inhibiting cell proliferation through controlling the cell cycle’s progression and triggering apoptosis [36]. Western blotting was used to measure the protein levels of the active phosphorylated form of p38 (p-p38). The findings indicate that MAE increased p-p38 levels in a concentration-dependent manner; following treatment with MAE, the cells increased by 1.3 ± 0.09-fold at 100 μg/mL and significantly by 1.5 ± 0.15-fold at 200 μg/mL (Figure 9B). Moreover, MAE significantly increased the levels of CDK inhibitors p21 and p27, which are downstream effectors of p38, to 1.61 ± 0.21 and 1.51 ± 0.26 at 200 μg/mL, respectively. In addition to being a downstream effector of p38 signaling, retinoblastoma protein (Rb) plays a part in differentiation, tumor suppression, cell cycle regulation, and apoptosis control [37]. P-rb levels were markedly decreased by MAE in a concentration-dependent manner (Figure 9B). Rb phosphorylation was significantly decreased by 200 μg/mL to 0.29 ± 0.1-fold compared to the vehicle control. The impact of MAE on cell cycle dynamics is further supported by these findings.

### 2.11. MAE Reduces iNOS and COX-2 Levels and Inhibits Angiogenesis In Ovo

The process by which tumor cells spread from the primary tumor site and metastasize into secondary sites results in the creation of new blood vessels to provide oxygen and nutrients to the cells. This process is known as angiogenesis [38]. The chick-embryo chorioallantoic membrane (CAM) assay was used to investigate how MAE influences angiogenesis. This was performed by applying MAE to the surface of the widely vascularized CAM over 24 h. By reducing the overall vessel length and number of junctions, MAE treatment of 200 µg/mL significantly suppressed angiogenesis, compared to the control, and resulted in a decrease in the number of junctions of 72.92 ± 3.44% and a reduction in the total vessel length of 44.56 ± 9.4% as seen in Figure 10B. Additionally, angiogenesis is mediated by a vital source of nitric oxide (NO), inducible nitric oxide synthase (iNOS) [39], and cyclooxygenase 2 (COX-2), an enzyme that produces prostaglandin E2 (PGE2) [40]. Our findings demonstrated that 200 µg/mL MAE significantly decreased the levels of COX-2 and iNOS proteins by 0.58 ± 0.038 and 0.52 ± 0.11, respectively, compared to the control. According to these findings, MAE suppresses angiogenesis by focusing on the synthesis of PGE2 and NO (Figure 10A).

## 3. Discussion

The development of medications for the prevention or treatment of illnesses continues to rely heavily on plants and their metabolites. Currently, there is growing interest in screening plants to identify therapeutic agents [41]. In this context, the Food and Drug Administration (FDA) has authorized three herbal mixture-based treatments, including antiallergenic, anticancer, and anti-psoriatic medications, in recent years [42]. However, *Mandragora autumnalis*’s medicinal potential has not been thoroughly studied. The novelty of the present study lies in its focus on elucidating the molecular mechanisms and chemopreventive pathways of MAE, specifically against triple-negative breast cancer MDA-MB-231 cells, a highly aggressive and treatment-resistant subtype. While previous studies mainly assessed the broad-spectrum cytotoxicity of *Mandragora autumnalis* extracts from flowers, fruits, whole plants, or crude leaf preparations against a variety of cancer cell lines, including hormone-receptor-positive breast cancer (MCF-7), lung cancer (A549), colon cancer (HCT-116), and murine mammary sarcoma (EMT6/p), these earlier studies primarily concentrated on general antiproliferative activity and reported results like decreased VEGF expression, low cytotoxicity toward normal cells, and tumor size reduction in animal models [19,43]. The present study, on the other hand, advances the field by combining phytochemical profiling with in-depth mechanistic analyses. It reveals that MAE modulates important cancer hallmarks, such as p53 activation, suppression of proliferation factor Ki67, inhibition of invasion and metastasis mediators MMP-9 and STAT-3, and alteration of cell cycle dynamics, in addition to exerting concentration and time-dependent cytotoxic effects. Our finding emphasizes that antioxidant pre-treatment using N-acetyl cysteine (NAC) could prevent ROS induced by MAE, highlight a ROS-mediated apoptosis pathway, and provide a level of mechanistic detail not found in previous studies. This study contributes to a better understanding of *Mandragora autumnalis*’s anticancer potential by advancing from descriptive cytotoxicity to molecularly targeted insights that are pertinent to the creation of novel TNBC treatments.

Numerous classes of phytochemical compounds, such as flavonoids, phenols, tannins, steroids, and essential oils, were found in MAE, according to the qualitative phytochemical analysis. Our findings concur with earlier research indicating the presence of coumarins and lipid-like compounds in extracts from *Mandragora* species [21]. As a result of its capacity to scavenge DPPH radicals, MAE demonstrated good antioxidant potential in vitro [44,45]. Moreover, phenols and flavonoids are recognized for their anticancer activity and anti-inflammatory and antioxidant functions [46]. The LC-MS profiling of the tested plant extract revealed a diverse set of bioactive phytochemicals, many of which have been documented in other medicinal plants for their promising anticancer properties, particularly against TNBC. TNBC is an aggressive breast cancer subtype lacking estrogen, progesterone, and human epidermal growth factor receptor 2 (HER2) receptors, often exhibiting poor prognosis and limited therapeutic options [47]. Hence, phytochemicals capable of multi-targeted anticancer action offer significant therapeutic value. In this study, the LC-MS analysis revealed a diverse profile of bioactive compounds, including alkaloids (hyoscyamine, tropine, tropinone, solacaproine), phenolic acids (chlorogenic acid), flavonoids (quercetin, chrysin), fatty acid derivatives (hexadecanamide, ethyl palmitate, linoleic acid, ethyl linolenate), and esters (ethyl hydrocinnamate, ethyl 3-hydroxybutanoate). Similar phytochemical compositions have been reported in medicinal plants like *Datura stramonium*, *Atropa belladonna*, and *Withania somnifera*, which are known for their rich alkaloid and flavonoid contents, targeted anticancer therapies. Among the major identified compounds, chlorogenic acid, scopoletin, caffeic acid, rutin, hyperoside, and linolenic acid have garnered substantial interest for their roles in inhibiting TNBC cell viability and metastasis. For instance, chlorogenic acid—widely found in *Coffea arabica*, *Lonicera japonica*, and *Solanum nigrum*—has demonstrated the ability to suppress TNBC cell proliferation, migration, and invasion through the modulation of key signaling pathways such as NF-κB and PI3K/Akt [48]. In MDA-MB-231 TNBC cells, chlorogenic acid downregulated MMP-9 expression and induced apoptosis, suggesting its chemopreventive potential [49]. Scopoletin, found in *Morinda citrifolia*, *Artemisia absinthium*, and *Scopolia carniolica*, appeared at high intensities in the sample, indicating a major role in bioactivity. This coumarin compound has been reported to inhibit TNBC cell growth by inducing oxidative stress-mediated apoptosis and enhancing chemosensitivity [50]. Rutin and hyperoside, both flavonol glycosides, are common in *Sophora japonica*, *Ginkgo biloba*, and *Hypericum perforatum* and exhibit anticancer potential by inhibiting epithelial–mesenchymal transition (EMT), a critical event in TNBC metastasis. Rutin has also shown synergistic effects with doxorubicin in reducing TNBC tumor burden [51]. Similarly, hyperoside inhibits the proliferation and migration of MDA-MB-231 cells by regulating miRNA expression and downregulating MMPs. Additionally, 3-methyl-2,5-furandione, a furan derivative, though less studied, shares structural similarity with compounds known for antiproliferative effects and may represent a novel candidate for anti-TNBC activity [52]. Collectively, the presence of these compounds, many of which are found in medicinal plants traditionally used in cancer therapy, supports the potential of this extract as a multi-targeted approach against cancer, especially TNBC.

Cancer is marked by uncontrolled proliferation, resulting from improper cell division and death regulation [53]. MAE was discovered to significantly inhibit the growth of MDA-MB-231 cells in a concentration-dependent manner. At the same time, Ki67, a highly expressed proliferation marker that is correlated with tumor severity, is significantly decreased [54]. Apoptosis induction is a useful tactic for limiting the growth of cancer cells, and apoptotic pathway targeting is becoming more and more crucial in the development of cancer treatments [55]. To better understand the anti-proliferative mechanism of the MAE extract, we studied apoptosis induction. MAE significantly inhibited MDA-MB-231 cell proliferation in a dose-dependent manner. Previous research has shown that *Mandragora autumnalis* inhibits the proliferation of various cancer cell lines [19]. This study is the first to evaluate the impact of MAE on the triple-negative subtype of breast cancer, including its underlying molecular mechanisms and cancer phenotypes.

Tumor growth and treatment failure are encouraged by TNBC’s frequent avoidance of apoptosis [56]. Consequently, activating apoptotic pathways is a highly effective method of treating cancer. The upregulation of the pro-apoptotic Bax protein and caspase-3, and the downregulation of pro-caspase 3 and the anti-apoptotic Bcl-2 protein, all indicated that MAE induced intrinsic apoptosis. This comes in line with previous studies on other plant extracts, including *Ziziphus nummularia* and *Haludule uninervis,* on triple-negative breast cancer cells [57,58].

Additionally, the p53 pathway was investigated to better understand the mechanism of MAE’s anti-proliferative effects. p53 is a tumor suppressor protein encoded by the *TP53* gene; its primary biological function appears to be to protect the DNA integrity of the cell. *TP53* has additional functions in development, aging, and cell differentiation [59,60]. Activated p53 promotes cell cycle arrest to allow DNA repair and apoptosis to prevent the spread of cells with severe DNA damage by transactivating target genes involved in cell cycle arrest and apoptosis [61]. Mutations in the p53 gene (*mtp53*) are associated with a variety of cancers, including 70–80% of TNBC [62]. It is noteworthy that the phosphorylation of Ser15 functions as a nucleation event that facilitates or encourages the sequential modification of numerous residues later on [63]. By preventing ubiquitylation, these alterations are believed to help stabilize p53 and imply that p53 regains its tumor-suppressive function [64]. MAE induced p53 phosphorylation in MDA-MB-231 cells, potentially restoring its wild-type conformation and increasing transcriptional activity. This suggests a promising therapeutic strategy for preventing cancer cell proliferation. Furthermore, the role of the p38 mitogen-activated protein kinase (MAPK) pathway was examined in order to obtain an understanding of the mechanism underlying the anti-proliferative effects of MAE. By controlling a number of cellular functions, such as cell cycle progression, apoptosis, and cellular stress response, p38 is known to be essential for preserving cellular homeostasis [65]. Our findings demonstrated that MAE raised p38 levels, which is consistent with other research demonstrating a link between p38 activation and the induction of cell cycle arrest [66]. Furthermore, MAE markedly elevated the levels of p27, a downstream effector of p38, and the cell cycle inhibitor protein p21. Moreover, MAE significantly reduced the phosphorylation of Rb, further implicating the p38 MAPK pathway in the proliferation of MDA-MB-231 cells. This can be explained by the activation of p38 MAPK, which stabilizes Cyclin-Dependent Kinase (CDK) inhibitors (e.g., p21 and p27), delaying Rb phosphorylation, which keeps Rb in its active, growth-suppressing form and causes cell cycle arrest in the G_1_ phase [67,68]. As a result of TNBC, increasing the levels of activity of p27, p21, p38, Rb, and p53 leads to a significant extension of the G_0_ and G_1_ phases of the cell cycle, preventing cancer cells from entering the DNA synthesis S phase. This cell cycle arrest is a desirable therapeutic effect, as it limits tumor cell proliferation and promotes apoptosis or senescence. Targeting this network of tumor suppressors and stress response proteins holds promise for developing more effective treatments for TNBC, which currently lacks targeted therapies.

Nearly all types of cancer have increased levels of ROS, which aid in the tumor’s growth and metastasis. A challenge for novel therapeutic strategies will be the delicate regulation of intracellular ROS signaling to effectively deprive cells of ROS-induced tumor-promoting events and tip the scales in favor of ROS-induced apoptotic signaling. On the other hand, therapeutic antioxidants may stop critical early stages of tumor development before ROS are produced [69]. The main intracellular antioxidant, glutathione, is directly derived from N-acetylcysteine (NAC). NAC is commonly used as a mechanistic probe in cancer research because it can reduce ROS accumulation and restore cell viability when a compound causes cancer cell death by generating reactive oxygen species (ROS) [70]. By using this method, our findings ascertained that oxidative stress plays a significant role in mediating the MAE’s cytotoxic effects. In that context, MAE simultaneously causes ROS production in MDA-MB-231 cells while exhibiting antioxidant activity in cell-free radical scavenging assays. The context- and dose-dependent redox characteristics of many phytochemicals, which can function as antioxidants in healthy physiological settings but have pro-oxidant, cytotoxic effects on cancer cells that are already under a lot of oxidative stress, are reflected in this dual behavior. On the other hand, a test like the 2′,7′-Dichlorodihydrofluorescein diacetate (DFCH-DA) would be useful to demonstrate that the extract does induce ROS production at the cellular level.

All of these points point to a scenario in which MAE causes ROS levels to rise, which in turn triggers anti-proliferative signaling pathways in MDA-MB-231 cells. It is possible that this signaling involves the ROS-p53 axis. Similar findings stated that when hysapubescin B, a withanolide derived from *Physalis pubescens*, was applied to HCT116 colorectal cancer cells, it produced ROS that inhibited mTORC1, which in turn activated autophagy and caused cell death [71].

Furthermore, we demonstrated that MAE impacted MDA-MB-231 cells’ ability to migrate and adhere to one another. These functions are essential to the epithelial–mesenchymal transition (EMT) and serve as a marker of the cancer’s spread to metastases. Cancer cells can migrate and invade more areas with less cell-to-cell adhesion [72].

According to the results of our investigation, treating MDA-MB-231 cells with MAE reversed the EMT phenotype and prevented metastasis by increasing the formation of cell–cell aggregates and decreasing their migration and invasion. The aggressive nature, poor prognosis, and treatment resistance of TNBC metastasis make it a significant clinical challenge [73]. Tumor cell invasion is commonly believed to require the epithelial-to-mesenchymal transition (EMT); however, mounting data suggest that there are other pathways through which tumor cells can spread. Using mesenchymal or amoeboid dissemination mechanisms, tumor cells can spread individually or in clusters. Mesenchymal cells move forward through the production of traction force through integrin-mediated extracellular matrix adhesion and cytoskeletal contractility [74]. Actin dynamics, integrin-based ECM adhesion, and proteolytic cleavage of ECM are the three factors that control this kind of tumor cell migration. The diverse tumor cells that migrate in cohesive groups preserve front-rear polarity and collaborate hierarchically [75]. Dysregulation of cell–cell adhesion and metalloproteinases’ degradation of the extracellular matrix (ECM) are two factors contributing to the rise in cancer cell migration and invasion [76]. The MDA-MB-231 cells’ ability to adhere to collagen was reduced by MAE treatment, suggesting that the cell-ECM interaction had been disrupted. Furthermore, MAE reduced the amounts of integrin β1, an adhesion molecule linked to TNBC’s heightened invasiveness and aggression [77]. These findings support MAE’s capacity to suppress TNBC cells’ capacity to spread by focusing on cell–cell and cell-ECM interactions. Boyden chamber and wound-healing assays were used in our study to examine the impact of MAE on the migration and invasion characteristics of MDA-MB-231. In this study, TNBC cell migration and invasion were markedly inhibited by MAE. Multiple investigations involving various plant extracts have demonstrated that the inhibition of MMP-2 and MMP-9 expression diminishes the migratory and invasive potential of MDA-MB-231 cells [78]. In our investigation, MAE significantly and concentration-dependently decreased the MDA-MB-231 cells’ cell–cell adhesion.

This was accompanied by a decrease in integrin β1 protein levels, suggesting that MAE could block tumor migration by inhibiting cell-to-cell adhesion. All things considered, MAE may be able to reduce TNBC metastasis by reducing cell-–ECM contact, preventing cell migration, adhesion, and invasion, and boosting cell–cell aggregation. More research is needed to determine whether the downregulation of MMPs mediates the MAE-induced inhibition of migration and invasion.

The necessity for more research is highlighted by the discovery of numerous signaling pathways implicated in the pathophysiology of cancer [79,80]. Despite progress in cancer research, providing more involved pathways and molecular targets is very important [81]. STAT3, in particular, plays important roles in a variety of cellular processes, including the cell cycle, cell proliferation, cellular apoptosis, and tumorigenesis [82]. Persistent activation of STAT3 has been reported in a variety of cancer types, and a poor prognosis of cancer may be associated with the phosphorylation level of STAT3 [83]. By controlling gene expression-related apoptosis, EMT, cell migration, aggregation, and invasion, STAT3 facilitates tumor growth and metastasis in TNBC [84]. STAT3 activation in the tumor microenvironment is regarded as an oncogenic event in addition to its typical cell functions. In cancer patients, elevated phospho-STAT3 expression is associated with a poor prognosis [85], and are strongly associated with cancer hallmarks and lead to poor patient outcomes [86].According to earlier research, substances derived from plants can lower cancer risk by blocking the STAT3 signaling pathway and the genes that are linked to it. For instance, cantharidin and baicalein both suppress the STAT3 pathway by downregulating EGFR and IL6 levels, respectively. Moreover, Hydroxy-jolkinolide B, Deoxy-2β,16-dihydroxynagilactone E, and Acetoxychavicol acetate inhibit triple negative breast cancer cells by downregulating of JAK/STAT pathway [86].

It has been demonstrated that phosphorylated STAT3 (p-STAT3), which was significantly decreased by MAE, downregulates important pro-metastatic factors like matrix metalloproteinase-9 (MMP-9), which was significantly decreased, effectively inhibiting tumor growth. When activated, the transcription factor p-STAT3 moves into the nucleus and stimulates the expression of genes related to invasion, metastasis, and cell survival. MMP-9, an enzyme that breaks down extracellular matrix components to promote cancer cell invasion and metastasis, is one of its well-established targets [87]. Thus, blocking STAT3 phosphorylation results in a decrease in MMP-9 expression [88], which in turn reduces TNBC cells’ capacity for invasion, which was achieved by MAE in this investigation. In order to reduce tumor aggressiveness and enhance outcomes for patients with TNBC, this regulatory axis emphasizes the therapeutic potential of targeting the STAT3/MMP-9 pathway.

Notably, MAE stimulates the production of ROS, suggesting that MAE contributes to oxidative stress in MDA cells. One of the main causes of DNA damage and a threat to genomic integrity is oxidative stress, which activates members of the p53 family, most notably p53, which is essential for tumor suppression [89]. The effects of MAE on p53, Ki-67, MMP-9, and STAT-3 are likely secondary to upstream molecular changes, particularly oxidative stress. The extract induces ROS production in MDA-MB-231 cells, and this effect is mitigated by the antioxidant NAC, suggesting that ROS generation is a critical mediator of the observed cellular responses. The induction of p53 phosphorylation following MAE treatment may represent a cellular response to oxidative DNA damage, leading to cell cycle arrest and apoptosis. Concurrently, the downregulation of Ki-67, MMP-9, and STAT-3 indicates a suppression of cell proliferation, invasion, and survival pathways, which are commonly regulated by p53 and other stress-responsive signaling mechanisms. Therefore, the modulation of these markers appears to be a consequence of ROS-induced signaling cascades rather than direct interactions with the extract components.

The development of new blood vessels, or angiogenesis, is an important step in the growth, invasion, and metastasis of tumors [90]. Particularly, TNBC has high angiogenic activity, which promotes fast tumor growth and makes metastasis easier. Therefore, blocking pro-angiogenic factors to target angiogenesis has become a promising therapeutic strategy for controlling the progression of TNBC [91]. Inducible nitric oxide synthase (iNOS) and cyclooxygenase-2 (COX-2) are two important pro-angiogenic enzymes that are upregulated in TNBC. In TNBC, their overexpression increases the risk of metastasis and vascularization [92]. Notably, treatment with MAE decreased the expression of COX-2 and iNOS in MDA-MB-231 cells. Additionally, in the chick embryo chorioallantoic membrane (CAM) assay, MAE significantly reduced vessel length and the number of junctions, demonstrating a strong anti-angiogenic effect in vivo-like environment. These results are similar to earlier studies that demonstrated that *Halodule uninervis*, a seagrass, also inhibits angiogenesis in TNBC [58].

To sum up, Effective therapeutic responses in TNBC are frequently characterized by downregulation of survival and invasion markers (e.g., Bcl-2, p-STAT3, MMP-9) and upregulation of pro-apoptotic and tumor suppressor proteins (e.g., Bax, Caspase-3, p21, p27, p-p53). Increases in E-cadherin and decreases in integrin/MMP-9 indicate that EMT and metastasis are suppressed. All markers studied in this investigation were listed with their change and proposed role after treatment with MAE (Figure 11).

Per these findings, MAE inhibited apoptosis, adhesion, migration, invasion, cell cycle, and angiogenesis in MDA-MB-231 cells. This study provides important insights into the anticancer potential of MAE, demonstrating its ability to reduce viability, modulate key molecular markers (p53, Ki-67, MMP-9, STAT-3), and affect multiple hallmarks of metastasis in MDA-MB-231 triple-negative breast cancer cells. The phytochemical composition and antioxidant properties of the extract were also characterized, providing a foundation for understanding its bioactivity. Despite these strengths, the study’s in vitro results show that MAE has anticancer potential against MDA-MB-231 cells; it lacks in vivo validation, which is necessary to evaluate the extract’s safety, pharmacokinetics, and efficacy in a setting that is more physiologically relevant. Mechanistic insight and reproducibility were also limited because, despite the extract’s phytochemical content and antioxidant qualities being described, no fractionation or identification of the precise bioactive compounds causing the observed effects was carried out. Also, the extract’s safety, effectiveness, and dosage in humans should be tested in the form of clinical trials. Moreover, a lack of precise quantitative analysis of the individual compounds in *Mandragora autumnalis* extract. While the TPC, TFC, and LC-MS data provide information on relative concentrations, they do not allow the determination of the exact amounts of bioactive compounds reaching the cellular level or whether these concentrations are physiologically achievable for therapeutic purposes. We acknowledge this important point and agree that precise quantification would provide clearer insight into the extract’s physiological relevance. Hence, to completely validate *Mandragora autumnalis*’s anticancer potential, these limitations highlight the necessity of in vivo research, the isolation of bioactive compounds, quantitative analyses, and clinical evaluation.

## 4. Materials and Methods

### 4.1. Collection of Mandragora Autumnalis Leaves and Preparation of Their Ethanolic Extract MAE

The leaves of *Mandragora autumnalis* were collected from south Lebanon in the spring. After being cleaned, the leaves were allowed to dry at room temperature. Mohammad Al-Zein, a resident plant taxonomist at the American University of Beirut (AUB) herbarium, identified the plants. A voucher specimen bearing the identification number GA 2025-1 is kept at the Post Herbarium, AUB. Leaves were washed and dried at room temperature and ground mechanically. For 72 h, the powder was shaken at 40 rpm in the dark while suspended in 80% ethanol. The suspension was then lyophilized using a freeze-dryer and filtered through filter paper. Dimethyl Sulfoxide (DMSO) was used to dissolve the resulting powder at a concentration of 100 mg/mL. The prepared plant extract MAE was then stored in the dark at 4 °C for further usage and analysis.

### 4.2. Phytochemical Analysis

The presence of primary and secondary metabolites was detected through qualitative tests to examine the chemical composition of the ethanolic and aqueous extracts of *Mandragora autumnalis*. Test for anthocyanins: A total of 0.5 g of the extract was dissolved in 5 mL of ethanol, followed by ultrasonication for 15 min at 30 °C. Subsequently, 1 mL of each extract was combined with 1 mL of NaOH (Sigma-Aldrich Co., St. Louis, MO, USA) and heated for 5 min at 100 °C. The presence of anthocyanins was indicated by the appearance of a bluish-green color. Anthraquinone test: 0.5 g of each extract (Sigma-Aldrich Co., St. Louis, MO, USA) was dissolved in 4 mL of benzene. 10% ammonia solution was added to the filtrate following filtration. The presence of anthraquinones was verified by the formation of a red or violet color. Test for cardiac glycosides: 5 mL of ethanol was used to dissolve 0.5 g of each extract. This was then ultrasonicated at 30 °C, filtered, and evaporated. Subsequently, the dehydrated extract was combined with 1 milliliter of glacial acetic acid (from Sigma-Aldrich Co., St. Louis, MO, USA) and a few drops of 2% FeCl_3_. Subsequently, the test tube’s side was filled with 1 milliliter of concentrated sulfuric acid (H_2_SO_4_) from Sigma-Aldrich Co., St. Louis, MO, USA. The appearance of a brown ring indicated the presence of cardiac glycosides. To test for essential oils, 0.5 g of each extract was dissolved in 5 mL of ethanol, then the mixture was ultrasonically sonicated at 30 °C and filtered. 100 µL of 1M NaOH was mixed with the filtrate. A tiny amount of 1M HCl (MERCK, Darmstadt, Germany) was then added. A white precipitate’s formation indicated the presence of essential oils. Test for flavonoids. One milliliter of 2% NaOH and 0.2 g of each extract were combined. When a concentrated, yellow-colored solution was achieved, a few drops of diluted acid were added to the mixture. The color in the solution vanished, indicating that flavonoids were present. Phenols test: 5 mL of ethanol was used to dissolve 0.5 g of each extract, which was then ultrasonically sonicated at 30 °C and filtered. A mixture of 2 mL distilled water and the filtrate was prepared. Subsequently, a small amount of 5% FeCl_3_ was added, and this caused a dark green color to form, indicating the presence of phenols. 5 mL of ethanol was used to dissolve 0.5 g of the extract, which was then ultrasonicated at 30 °C and filtered. The filtrate was then mixed with 1 mL of concentrated sulfuric acid (H_2_SO_4_). The development of a red hue was indicative of the presence of quinones. To test for tannins, dissolve 0.5 g of each extract in 5 mL of distilled water, then filter and ultrasonicate at 80 °C. Once the filtrate had cooled to room temperature, five drops of 0.1% FeCl_3_ were added. A brownish-green or blue-green coloration suggested the presence of tannins. To test for terpenoids, dissolve 0.5 g of the extract in 5 mL of chloroform, then filter and ultrasonicate at 30 °C. The filtrate was then mixed with 2 mL of concentrated sulfuric acid (H_2_SO_4_). The creation of a reddish-brown hue indicated the presence of quinones.

### 4.3. Total Phenolic Content (TPC)

With a few minor modifications, the Folin–Ciocalteu method was used to determine the total polyphenol content (TPC) of MAE [93]. The concentration of both extracts was made to be 1 mg/mL. 500 μL of the extract was separated, combined with 2.5 mL of Folin–Ciocalteu reagent, and left to oxidize for five minutes. After adding 2 mL of a 75 g/L sodium carbonate solution to neutralize the reaction, the mixture of the extract was left to incubate for 1 h at 37° C in the dark. Following incubation, the samples’ absorbance at 765 nm was compared to standards of gallic acid, which is a known polyphenol. The TPC of the extract was given as a percentage (mg GAE/g) of the total gallic acid equivalents per gram of dry leaves used in the extract’s preparation. The TPC analysis was carried out three times, and the mean values ± SEM of the outcomes are shown.

### 4.4. Total Flavonoid Content (TFC)

Using a modified aluminum chloride colorimetric assay, the total flavonoid content (TFC) of MAE was ascertained [93]. To put it briefly, a concentration of 1 mg/mL was used to prepare the extract. Then, an aliquot (0.5 mL) of the extract was combined with 1.5 mL of 95% ethanol, 2.8 mL of ultra-pure distilled water, 0.1 mL of a 10% methanolic aluminum chloride solution, and 0.1 mL sodium acetate. Using quercetin as a standard, the absorbance was measured at 415 nm following a 30 min dark incubation period at room temperature. The TFC was measured in (mg QE/g) of quercetin equivalents per gram of dry leaves used to prepare the extract. Three runs of this analysis were conducted, and the results are shown as mean values ± SEM.

### 4.5. The Antioxidant Activity (DPPH) of Mandragora Autumnalis Ethanolic Extract

Using the free-radical-scavenging activity of α, α-diphenyl-α-picrylhydrazyl (DPPH), the antioxidant activity of MAE was assessed. MAE at different concentrations (50, 100, 200, 400, or 600 μg/mL) was combined with a 0.5 mM DPPH solution in methanol. DPPH solution (0.5 mL), methanol (3 mL), and 80% ethanol (0.5 mL) made up the blank solution, which was used for comparison. The combined samples were then exposed to darkness for 30 min, and a spectrophotometer was used to measure the optical density (OD) at a wavelength of 517 nm. The following formula was used to calculate the percentage of DPPH-scavenging activity for each MAE concentration. For comparison, ascorbic acid was used as the standard. Percentage of inhibition (absorbance of control − absorbance of the extract)/(absorbance of control) × 100.

### 4.6. Liquid Chromatography-Mass Spectrometry

The Stock solution was prepared by dissolving 5 mg of the MAE sample in a solution of 50 µL of DMSO and 450 µL of methanol, then 250 µL of the solution was diluted with 1 mL of methanol and then used for identification of the metabolite in the LC-MS system. All the other reagents used, Acetonitrile, methanol, water, and formic acid, were LC/MS grade. A Bruker Daltonik (Bremen, Germany) Impact II ESI-Q-TOF System equipped with Bruker Dalotonik. The Elute UPLC system (Bremen, Germany) was used for screening compounds of interest. We used standards for identification of *m*/*z* with high-resolution Bruker TOF MS and exact retention time of each analyte after chromatographic separation. This instrument was operated using the Ion Source Apollo II ion Funnel electrospray source. The capillary voltage was 2500 V, the nebulizer gas was 2.0 bar, the dry gas (nitrogen) flow was 8 L/min, and the dry temperature was 200 °C. The mass accuracy was ˂1 ppm; the mass resolution was 50,000 FSR (Full Sensitivity Resolution), and the TOF repetition rate was up to 20 kHz. using Elute UHPLC coupled to a Bruker Impact II QTOFMS. Chromatographic separation was performed using Bruker Solo 2.0_C-18 UHPLC column (100 mm × 2.1 mm × 2.0 μm) at a flow rate of 0.51 mL/min and a column temperature of 40 °C. Solvents: (A) water with 0.1% methanol and (B) Methanol injection volume 3 mL. Data analysis was performed using Bruker’s MetaboScape version 5.0 and DataAnalysis software version 5.1. Compound annotation in MetaboScape was conducted using the MetaboBase library, a list of standards, as well as a custom list of previously reported compounds specific to the plant under investigation. The raw data files were converted to the mzML format using msConvert version 3.0, a C++ tool from the ProteoWizard suite designed to convert raw files into standard mass spectrometry formats. The resulting mzML files were analyzed using an in-house R script based on the xcms version 1.54.0, CAMERA version 1.34.0, and MSnbase version 2.8.3 packages. Chemical formulas for annotated ions were generated and checked using the Rdisop package version 2.13.0. The isotopic pattern for the annotated chemical formulas matched the theoretical pattern with an accuracy of more than 99%. Chromatograms and spectra were generated using both Bruker Data Analysis and the ggplot2 library in R version 3.5.2. The analyses performed using the two methods yielded comparable results.

### 4.7. Cell Culture

Human breast cancer cells MDA-MB-231 were acquired from the American Type Culture Collection (Manassas, VA, USA). The cells were cultured in a 37 °C, 5% CO_2_ humidified chamber using DMEM high-glucose medium supplemented with 10% fetal bovine serum (FBS; Sigma-Aldrich, St. Louis, MO, USA) and 1% penicillin/streptomycin (Corning, MA, USA).

### 4.8. MTT Cell Availability Assay

MDA-MB-231 cells were seeded in a 96-well tissue culture plate at a density of 5 × 10^3^ cells/well, and they were left to grow until 30% confluence was reached. The cells were subjected to escalating concentrations of MAE (50, 100, 200, 400, 600, and 1000 μg/mL) for 72 h. The viability of the cells was assessed using 3-(4,5-dimethylthiazol-2-yl)-2,5-diphenyltetrazolium bromide (MTT; Sigma-Aldrich, St. Louis, MO, USA). The viability of the cells was determined by comparing the proportionate viability of the treated cells to the vehicle-treated cells (equivalent concentration of DMSO), whose viability was assumed to be 100%.

### 4.9. Migration (Scratch) Assay

In 12-well plates, MDA-MB-231 cells were cultured until a confluent cell monolayer was formed. Next, a 10 μL pipette tip was used to create a scratch across the confluent cell monolayer. After removing the cell culture medium, the cells were cleaned with PBS to get rid of any debris. Following the addition of fresh medium containing MAE at the designated concentration (100, or 200 μg/mL), cells were incubated at 37 °C. With an inverted microscope (objective ×4), photomicrographs of the scratch were taken at baseline (0 h) and 10 h post-scratch. Using the ZEN software (Zeiss, Jena, Germany), the width of the scratch was measured and expressed as the average difference ± SEM between the measurements made at time zero and the specified time point (10 h).

### 4.10. Trans-Well Migration Assay

To test MDA-MB-231 cells’ migratory potential, trans-well inserts (8 µm pore size; BD Biosciences, Bedford, MA, USA) were employed. 3 × 10^5^ cells were seeded into the insert’s upper chamber. After that, cells were given the indicated MAE concentrations or not. A DMEM medium with 10% FBS was added as a chemoattractant in the lower cer. After that, the cells were cultured at 37 °C and given 24 h to migrate. The cells on the insert’s upper surface were eliminated using a sterile cotton swab. Additionally, cells that moved to the insert’s lower surface were stained with DAPI (1 µg/mL) and fixed with 4% formaldehyde. and examined for quantification at a 10× magnification using a fluorescence microscope. Three assay runs were conducted, and the results are shown as mean values ± SEM.

### 4.11. Matrigel Invasion Assay

To evaluate MDA-MB-231 cells’ invasive potential, a BD Matrigel Invasion Chamber (8 µm pore size; BD Biosciences, Bedford, MA, USA) was utilized. The experiment is comparable to the trans-well migration chamber assay, except that a diluted 1:20 Matrigel matrix is added. For quantification, cells that penetrated the Matrigel layer to the insert’s lower surface were fixed with 4% formaldehyde, stained with DAPI, and examined under a fluorescence microscope. Three assay runs were conducted, and the results are shown as mean values ± SEM.

### 4.12. Aggregation Assay

MDA-MB-231 cells were collected from confluent plates using sterile 2 mM EDTA in Ca^2+/^Mg^2+^-free PBS to measure cell aggregation. The cells were divided into individual non-adherent culture plates and subjected to MAE treatment (100 or 200 μg/mL). After three hours of gentle shaking at 90 rpm and 37 °C, the cells were fixed with 1% formaldehyde. Photomicrographs were taken to be examined using an Olympus IX 71 (Hachioji, Tokyo, Japan) inverted microscope.

### 4.13. Adhesion Assay

For a duration of 24 h, MDA-MB-231 cells were cultured with or without different concentrations (100 or 200 μg/mL) of MAE. Following a collagen pre-coating, they were seeded onto 24-well plates and incubated for 60 min at 37 °C. The MTT reduction assay was then used to determine the quantity of adherent cells after the cells had been washed with PBS to remove non-adherent cells.

### 4.14. Analysis of Apoptotic Morphological Changes

Using a phase-contrast inverted microscope, characteristics associated with apoptotic cells were observed. For this, different concentrations of MAE (100 or 200 μg/mL) were either present or absent when growing cells in 6-well plates. After 24 h, images at 4×, 10×, and 20× magnifications were captured. 4′, 6-diamidino-2-phenylindole, dihydrochloride (DAPI) staining was used to identify changes in nuclear morphology indicative of apoptosis. For a full day, MDA-MB-231 cells were cultured in a 12-well plate with or without the indicated concentrations of MAE (100 or 200 μg/mL). Following the manufacturer’s instructions, the cells were stained with DAPI (Cell Signaling #4083), fixed with 4% formaldehyde, and then fluorescence microscopy was used to visualize the apoptotic morphological alterations.

### 4.15. Western Blot Analysis

To create whole-cell lysates, MDA-MB-231 cells were subjected to two PBS washes before being lysed in a lysis buffer that contained 60 mM Tris and 2% SDS (pH of 6.8). After that, the lysate mixture was centrifuged for 10 min at 1.5 × 104 g. Bradford protein quantification kit (Biorad, Hercules, CA, USA) was used to measure the amount of protein in the resultant supernatant. Protein extracts aliquots ranging from 25 to 30 μg were resolved using 10% sodium dodecyl sulfate-polyacrylamide gel electrophoresis (SDS-PAGE) and subsequently placed onto an Immobilon PVDF membrane (Biorad).

PVDF membrane was blocked with a 5% non-fat dry milk solution in TBST (Tris-buffered saline with 0.05% Tween 20) for one hour at room temperature. To perform immunodetection, a particular primary antibody was incubated with the PVDF membrane for an entire night at 4 °C. After removing the primary antibody and washing the membrane with TBST, the membrane was incubated for an hour with the secondary antibody, horseradish peroxidase-conjugated anti-IgG. Thermo Scientific, Rockford, IL, USA) supplied an enhanced chemiluminescence (ECL) substrate kit to visualize immunoreactive bands after washing the secondary antibody with TBST, by the manufacturer’s instructions. Sourced from Cell Signaling (Cell Signaling Technology, Inc., Danvers, MA, USA), all primary and secondary antibodies were used.

### 4.16. Gelatin Zymography

In a 100 mm tissue culture plate, MDA-MB-231 cells (1.0 × 10^6^) were cultivated in serum-free DMEM medium with or without varying concentrations (100, or 200 μg/mL) of MAE. The cultures’ conditioned media were gathered and concentrated following a 24 h incubation period. A 10% non-reducing polyacrylamide gel containing 0.1% gelatin was used to separate 30 µg of proteins. After electrophoresis, the gels were rinsed for one hour in 2.5% (*v*/*v*) Triton X-100 to get rid of SDS. They were then incubated for the entire night at 37 °C in a solution that contained 50 mM Tris-HCl (pH 7.5), 150 mM NaCl, 0.5 mM ZnCl_2_, and 10 mM CaCl_2_ to enable the media’s proteases to break down the gelatin substrate enzymatically. A 0.5% solution of Coomassie brilliant blue R-250 was used to stain the resultant gel. The gel showed clear patches that showed matrix metalloproteinases (MMPs) breaking down the gelatin. Using ImageJ software version 1.54p, densitometry analysis was performed, and each cleared band’s density was normalized to a nonspecific band on the gel that was equally loaded.

### 4.17. Flow Cytometry Analysis of Cell Cycle

MDA-MB-231 cells were cultivated in 100 mm tissue culture plates before incubating with or without MAE. Cells were extracted and then suspended in 500 µL of PBS after 24 h. After that, the cells were fixed using an equivalent volume of 100% ethanol and maintained at −20 °C for at least 12 h. The cells were then centrifuged, pelleted, and washed with PBS before being resuspended in PBS containing 1 µg/mL of 4′, 6-diamidino-2-phenylindole dihydrochloride (DAPI, Cell Signaling Technology, Inc., Danvers, MA, USA) and incubated at room temperature for half an hour. After that, the cell samples were evaluated using the Becton Dickinson BD FACSCanto II Flow Cytometry System (Franklin Lakes, NJ, USA), and FACSDiva 6.1 software was used to facilitate data.

### 4.18. Chorioallantoic Membrane

Fertilized chicken eggs were cleaned with 70% ethanol, rotated, and incubated at 37 °C and 60% relative humidity. After a week, the air sac was revealed when the highly vascularized chorioallantoic membrane (CAM) was dropped by making an opening in the eggshell. The CAM was treated with 200 μg/mL of MAE and incubated for 24 h to investigate its impact on blood vessel growth between treatment and control. AngioTool 0.5 software was then used to measure the lengths of the vessels and count the number of junctions in the CAM images that were taken.

### 4.19. Statistical Analysis

The results were analyzed using the student *t*-test. When comparing more than two means, either one-way ANOVA followed by Dunnett’s post hoc test or two-way ANOVA followed by Tukey–Kramer’s post hoc test were used. *p*-values less than 0.05 were considered statistically significant.

## 5. Conclusions

Overall, this study showed that *Mandragora autumnalis* ethanolic leaf extract is a promising phytopharmacological candidate for the treatment of triple-negative breast cancer, a highly aggressive and therapeutically refractory breast cancer subtype. The observed decrease in MDA-MB-231 cell viability in a dose- and time-dependent manner reflects the cytotoxic potential of MAE, aligning with previous studies on other Solanaceae species known for their bioactive alkaloids and phenolics. Notably, the significant upregulation of the tumor suppressor p53 and concomitant downregulation of proliferation and metastasis-associated markers (Ki67, MMP-9, STAT-3) reinforce the mechanistic plausibility of MAE-induced tumor suppression. These molecular events are consistent with canonical antitumor pathways modulated by natural agents. The capacity of MAE to modulate these pathways points to its potential for not only cytotoxic but also anti-metastatic and anti-angiogenic applications. Furthermore, the interference of MAE with critical metastatic processes—including cell migration, invasion, angiogenesis, aggregation, and adhesion—broadens its functional relevance. These phenotypic suppressions reflect anti-metastatic activity comparable to compounds like quercetin and apigenin. Unlike many monotherapeutic agents that narrowly target single pathways, the broad-spectrum bioactivity of MAE implies a synergistic interplay of its phytochemical constituents, which is characteristic of botanical therapeutics and presents an advantage in overcoming the molecular heterogeneity and drug resistance typical of TNBC.

## Figures and Tables

**Figure 1 ijms-26-08506-f001:**
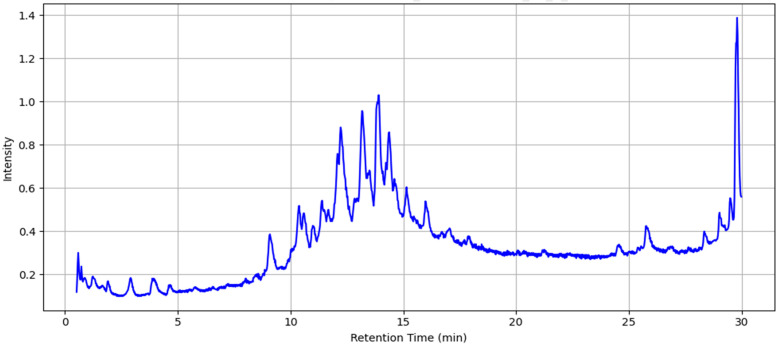
Total ion chromatogram of MAE obtained using LC-MS (the intensity is multiplied by 10^7^ a.u.).

**Figure 2 ijms-26-08506-f002:**
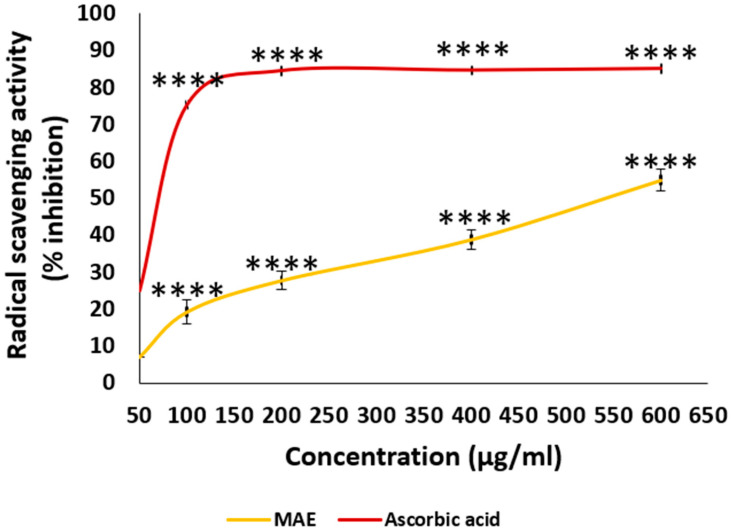
The antioxidant capacities of various concentrations of MAE (50, 100, 200, 400, and 600 µg/mL) were measured using the DPPH free-radical-scavenging assay. The reference used was ascorbic acid. The values are displayed as the three experiments’ average ± standard error (**** *p* < 0.0001).

**Figure 3 ijms-26-08506-f003:**
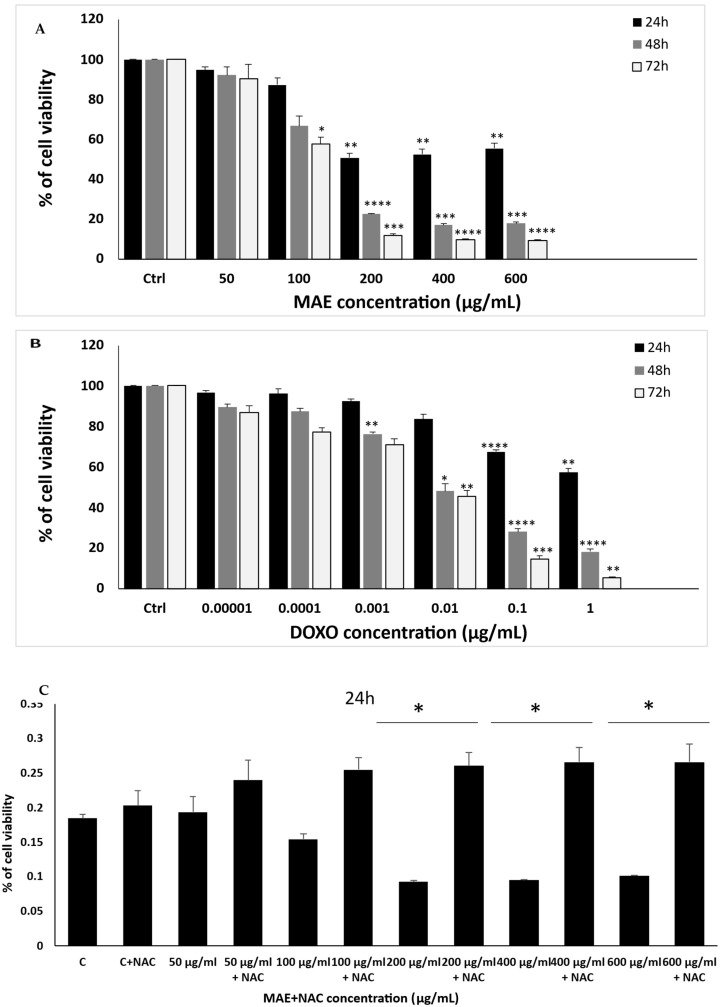
The cellular proliferation of MDA-MB-231 cancer cells is inhibited by MAE. (**A**) MDA-MB-231 cells received the indicated MAE concentrations for 24, 48, and 72 h. The MTT assay, detailed in Materials and Methods, was used to assess the viability of the cells. The information is presented as a percentage and shows the mean ± SEM of three separate triplicate experiments (*n* = 3), comparable to control cells. (**B**) After 24, 48, and 72 h of treatment, cell viability was assessed using either the vehicle control or the specified DOXO concentrations. (**C**) Before being treated with MAE for 24 h, MDA-MB-231 cells were pre-treated with NAC (10 mM) for 30 min. Values are shown as the mean ± SEM of three separate experiments and are expressed as a percentage of the vehicle control. (**D**) The indicated concentrations of MAE (100 and 200 μg/mL) were incubated with and without MDA-MB-231 cells for 24 h. After the cells were lysed, protein lysates were loaded with β-actin and subjected to Western blotting using a Ki67 antibody. The data are presented as a percentage of the corresponding control cells and show the mean ± SEM of three experiments (*n* = 3). One-way ANOVA was used for statistical analysis, and the LSD post hoc test was used after (* *p* < 0.05, ** *p* < 0.01, *** *p* < 0.001, and **** *p* < 0.0001).

**Figure 4 ijms-26-08506-f004:**
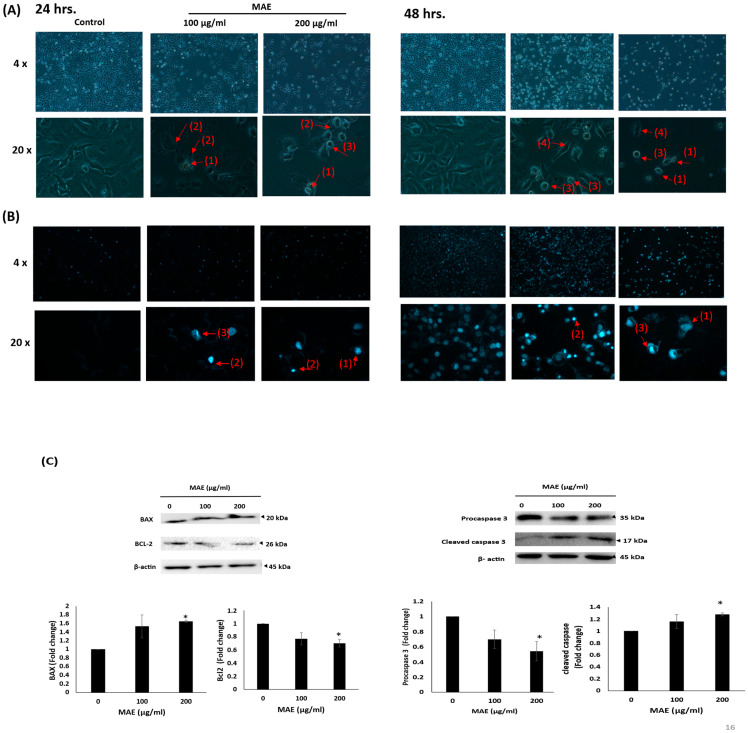
(**A**) Changes in the morphology of cells were observed using light optical microscopy. (1) Apoptotic bodies, (2) echinoid spikes, (3) membrane blebbing, and (4) cell shrinkage are all indicated by arrows. For 24 and 48 h, cells were treated with either a vehicle-containing control or MAE (100 or 200 µg/mL). (**B**) After that, they were stained with DAPI so that fluorescence microscopy could be used to see changes in nuclear morphology. Photos were captured with a 10× magnification. Arrows depict (1) chromatin lysis, (2) nuclear condensation, and (3) apoptotic bodies. (**C**) MAE (100 or 200 µg/mL) or a vehicle-containing control was used to treat the cells. Using Western blotting, the protein levels of Procaspase-3, cleaved Caspase 3, Bcl-2, and Bax were ascertained. Loading control was established using β-actin. The three separate experiments’ mean ± SEM (*n* = 3) is represented by the data. (* indicates *p* < 0.05).

**Figure 5 ijms-26-08506-f005:**
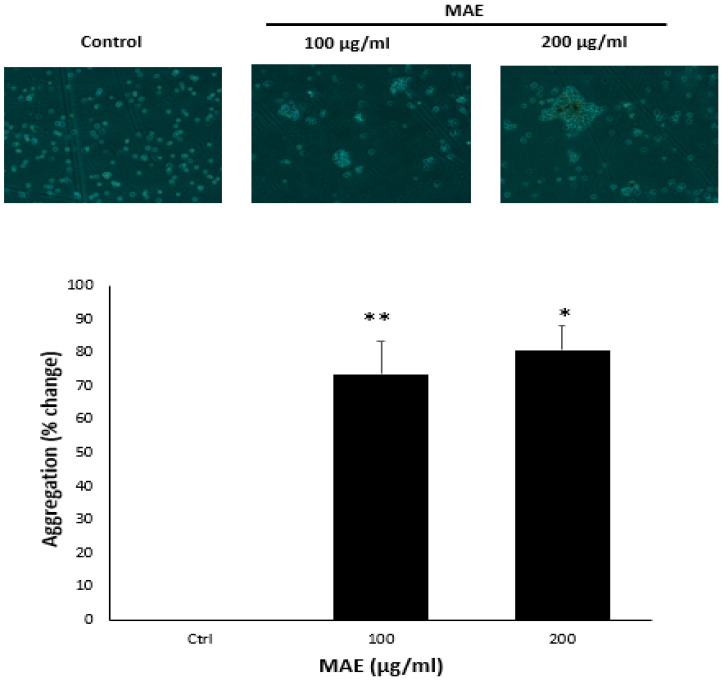
MAE promotes intercellular aggregation and E-cadherin levels in MDA-MB-231 breast cancer cells. After incubating MDA-MB-231 cells with either 100 or 200 µg/mL of MAE or a vehicle-containing control, the cells were tested for cell aggregation. The cells were photographed at a magnification of 4× after 4 h, and the percentage of cell–cell aggregates was calculated as mentioned in the Materials and Methods section. Western blotting was used to analyze the protein levels of E-cadherin, using β-actin as a loading control, where MAE (100 or 200 µg/mL) or a vehicle-containing control was applied to MDA-MB-231 cells for 24 h. The three separate experiments’ mean ± SEM (*n* = 3) is represented by the data (* *p* < 0.05, ** *p* < 0.005).

**Figure 6 ijms-26-08506-f006:**
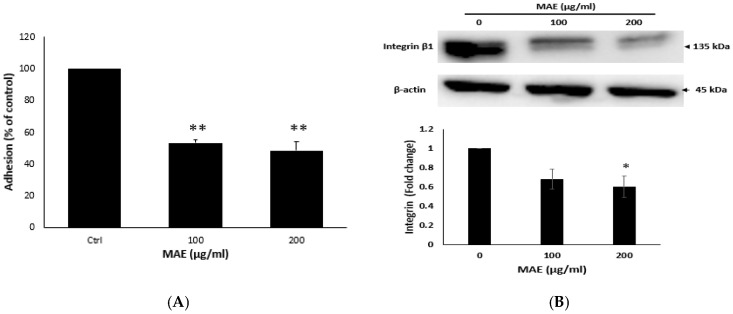
MAE inhibits the adherence of MDA-MB-231 breast cancer cells to collagen and lowers the levels of the protein integrin β1. (**A**) MDA-MB-231 cells were cultured for 24 h with either MAE (100 or 200 µg/mL) or a vehicle-containing control. After that, cells were seeded into wells coated with collagen and given three hours to adhere. The MTT assay was used to measure adhesion, and the results were reported as a percentage of the corresponding control cells. (**B**) For 24 h, MDA-MB-231 cells were cultured with either vehicle-containing control or MAE (100 or 200 µg/mL). β-actin served as a loading control when integrin β1 expression levels in whole-cell protein lysates were examined by Western blotting. The data show the average ± standard error of three separate experiments (*n* = 3), (* *p* < 0.05, ** *p* < 0.005).

**Figure 7 ijms-26-08506-f007:**
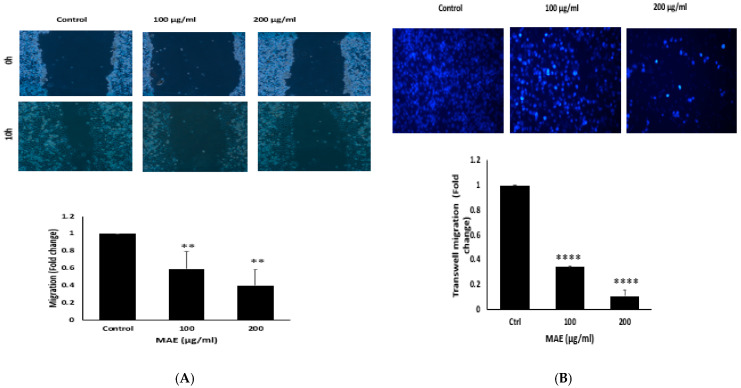
Using an inverted phase-contrast microscope at the designated time points and 4× magnification, photomicrographs of the wound were taken. (**A**) Plotted values represent the fold change in migration relative to control cells treated with a vehicle. (**B**) In Boyden chamber trans-well inserts, MDA-MB-231 cells were treated overnight with either MAE (100 or 200 µg/mL) or a vehicle-containing control. After migrating to the chamber’s lower surface, the cells were counted, examined, photographed at a 4× magnification, and stained with DAPI. The data show the average ± standard error of three separate experiments (*n* = 3). (** *p* < 0.005, **** *p* < 0.0001).

**Figure 8 ijms-26-08506-f008:**
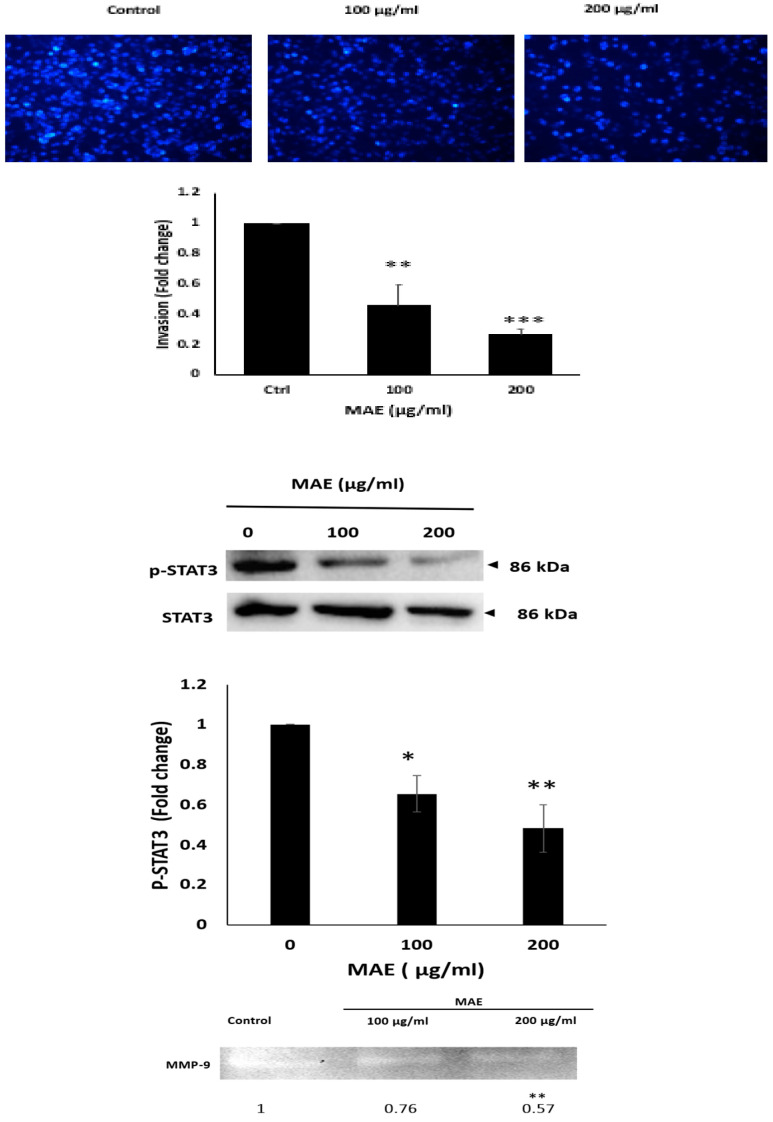
The invasive potential of MDA-MB-231 breast cancer cells is decreased by MAE. For 24 h, MDA-MB-231 cells were cultured in Boyden chamber trans-well inserts that had been previously coated with Matrigel, either with a vehicle-containing control or with 100 or 200 µg/mL of MAE. The Matrigel layer’s invading cells were imaged, counted, and examined using DAPI staining. Images were captured with a 4× magnification. The three separate experiments’ mean ± SEM (*n* = 3) is represented by the data. (** *p* < 0.005 and *** *p* < 0.001). Western blotting was used to measure the phosphorylated STAT3 protein levels, with β-actin serving as a loading control. Cells were treated either with a vehicle-containing control or with MAE (100 or 200 µg/mL). The results showed that the STAT3 signaling pathway is inhibited by MAE. Data represent the mean ± SEM of three independent experiments (*n* = 3), (* *p* < 0.05 and ** *p* < 0.005). After being seeded in serum-free media, MDA-MB-231 cells were treated with 100 or 200 µg/mL of MAE. Then the conditioned media were concentrated and put through gelatin zymography to determine MMP-9 activity (* *p* < 0.05 and ** *p* < 0.01).

**Figure 9 ijms-26-08506-f009:**
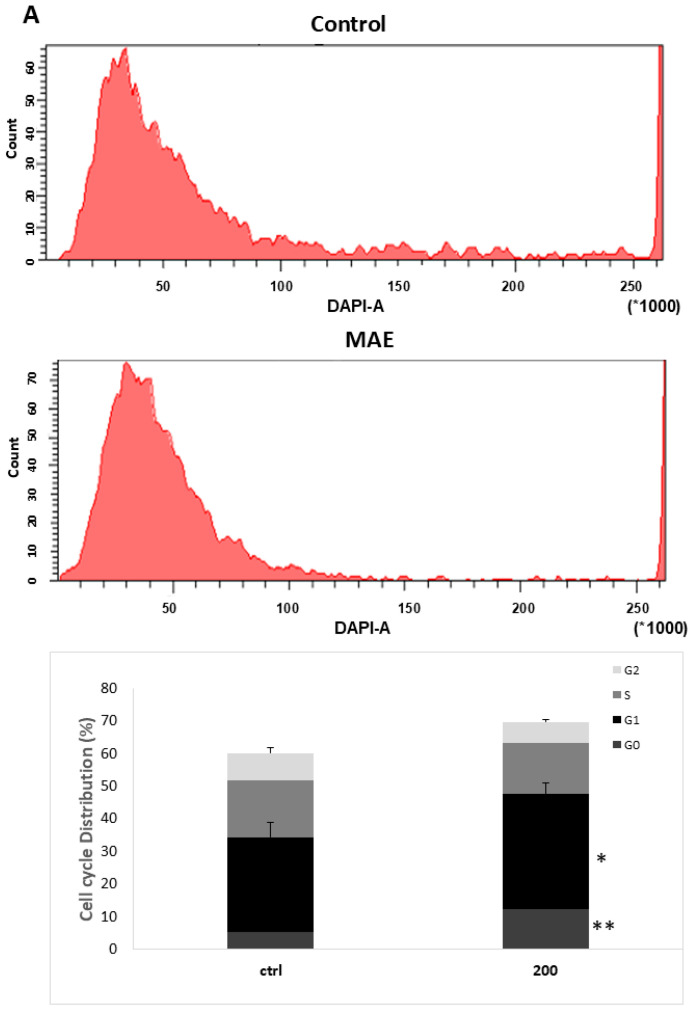
The G_0_/G_1_ cell cycle arrest in MDA-MB-231 cells is induced by MAE. (**A**) MDA-MB-231 cells were cultured for 24 h with 200 µg/mL of MAE or a vehicle-containing control. Following collection and fixation, cells were stained with 4′, 6-diamino-2-phenylindole (DAPI) and subjected to flow cytometry analysis. The mean ± SEM of three separate experiments (*n* = 3) is represented by the data. (**B**) MAE (200 µg/mL) or a vehicle-containing control was used to treat the cells. Western blotting was used to measure the amounts of phosphorylated p38, p21, p27, Rb, and phosphorylated p53 proteins. To control loading, β-actin was employed. The mean ± SEM of three separate experiments (*n* = 3) is represented by the data. ** indicates *p* < 0.005, and * indicates *p* < 0.05.

**Figure 10 ijms-26-08506-f010:**
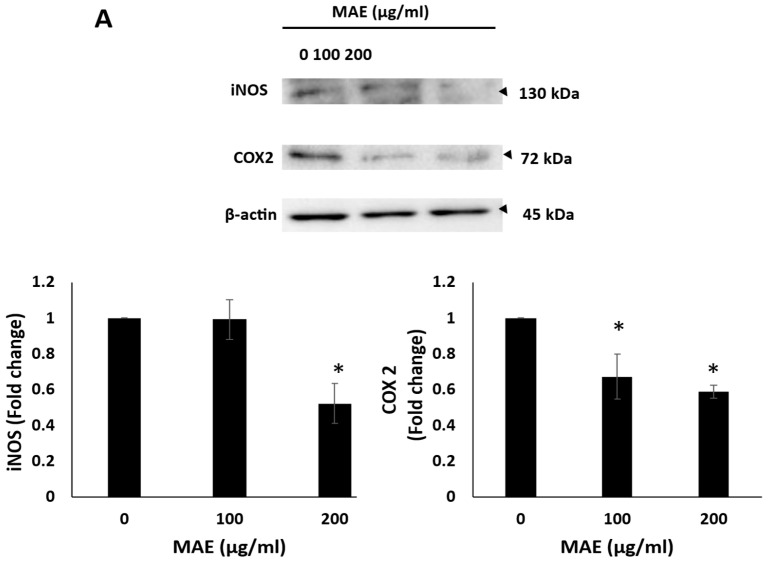
MAE extract lowers the amounts of iNOS and COX-2 in MDA-MB-231 breast cancer cells and suppresses angiogenesis in fertilized chicken eggs. (**A**) MAE lowers the amounts of iNOS and COX-2 in MDA-MB-231 breast cancer cells and prevents angiogenesis in fertilized chicken eggs. (**B**) For 24 h, MAE was applied to the fertilized chicken eggs’ chorioallantoic membranes (CAMs). Pictures were taken 24 h before the fertilized chicken eggs’ chorioallantoic membranes (CAMs). To score the angiogenic response, pictures were taken both before and after treatment. AngiTool 0.5 is the software that was used to measure the total length of the vessel and the total number of junctions. Western blotting was used to measure the protein levels of COX-2 and iNOS, with β-actin serving as a loading control. The results were displayed as the percentage change compared to the vehicle-treated control. The mean ± SEM of three separate experiments (*n* = 3) is represented by the data (* *p* < 0.05).

**Figure 11 ijms-26-08506-f011:**
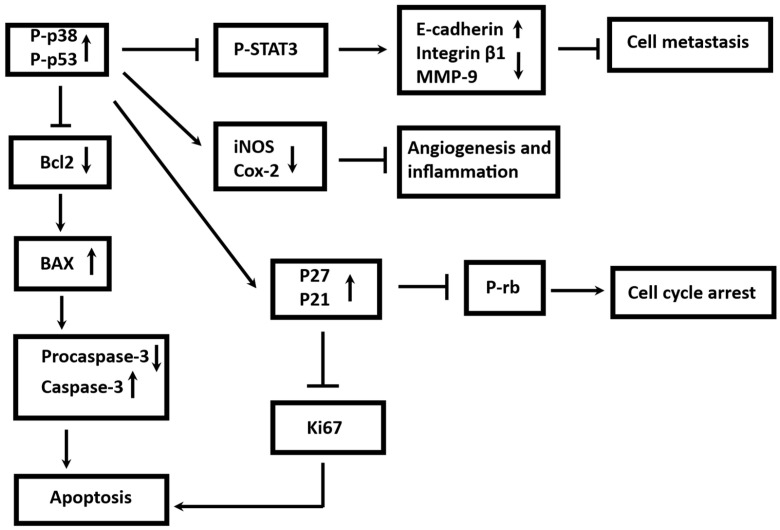
The proposed mechanism of action of MAE in the treatment of TNBC through the mediation of the selected markers →: triggers, 
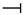
: inhibits.

**Table 1 ijms-26-08506-t001:** Qualitative Phytochemical analysis of *Mandragora autumnalis* ethanolic extract (MAE), (+) indicates the presence and (-) indicates the absence of the secondary metabolite.

Metabolite	MAE
Anthraquinones	-
Tannins	+
Resins	-
Terpenoids	+
Flavonoids	+
Quinones	-
Anthocyanins	-
Saponins	-
Phenols	+
Steroids	+
Cardiac glycosides	-
Fixed oils and fatty acids	+

**Table 2 ijms-26-08506-t002:** TPC and TFC of *Mandragora autumnalis* ethanolic extract.

Assay Type	MAE
TPC (mg GAE/g)	58.98 ± 7.40
TFC (mg QE/g)	36.47 ± 0.87

**Table 3 ijms-26-08506-t003:** Identified compounds in the ethanolic extract of *Mandrgaora autumnalis* using both (A) positive and (B) negative ionization modes in LC-MS.

A. Positive Ionization Mode						
Number	*m*/*z*	RT [min]	Ions	Compound Name	Molecular Formula	Intensity
1	127.0389	0.58	[M + H]+	5-Hydroxymethyl-2-furancarboxaldehyde	C_6_H_6_O_3_	49,130.136
2	133.0827	0.71	[M + H]+	Ethyl 3-hydroxy-butanoate	C_6_H_12_O_3_	72,955.974
3	140.1066	0.86	[M + H]+	Tropinone	C_8_H_13_NO	21,053.859
4	149.0596	1.23	[M + H]+	3-(Methylthio)propyl acetate	C_6_H_12_O_2_S	8790.512
5	117.0542	1.3	[M + H]+	1-Hydroxy-2-propanone acetate	C_5_H_8_O_3_	8646.324
6	193.0492	2.91	[M + H-C_6_H_10_O_5_]+	Chlorogenic acid	C_16_H_18_O_9_	669,015.883
	355.1018		[M + H]+			221,652.966
	445.0708		[M + Na + NaCOOH]+			20,681.809
7	619.2479	2.91	[M + H]+	Simulanoquinoline	C_37_H_34_N_2_O_7_	11,326.110
8	641.2302		[M + Na]+			90,245.22
9	290.1745	3.89	[M + H]+	Hyoscyamine	C_17_H_23_NO_3_	5,938,808.003
10	303.0494	9.16	[M + H]+	Quercetin	C_15_H_10_O_7_	20,868.344
11	179.1178	9.6	[M + H]+	Ethyl hydrocinnamate	C_11_H_14_O_2_	24,858.923
12	255.0862	13.43	[M + H]+	Chrysin	C_15_H_10_O_4_	11,043.2
13	281.266	26.76	[M + H]+	Linoleic acid	C_18_H_32_O_2_	70,427.883
14	311.2933	27.7	[M + H]+	Ethyl oleate	C_20_H_38_O_2_	4151.762
15	243.2505	28.62	[M + H]+	n-Pentadecanoic acid	C_15_H_30_O_2_	13,652.643
16	193.1581	29.04	[M + H]+	Ionone (β-Ionone)	C_13_H_20_O	10,667.398
17	307.266	29.32	[M + H]+	Ethyl linolenate	C_20_H_34_O_2_	10,224.308
18	156.138	29.41	[M + H]+	Methylisopelletierine	C_9_H_17_NO	48,479.510
19	114.0911	29.41	[M + H-C_2_H_4_]+	Tropine	C_8_H_15_NO	42,742.76
	142.1224		[M + H]+			57,239.581
20	336.2868	29.49	[M + Na]+	Solacaproine	C_18_H_39_N_3_O	8765.329
21	256.2629	29.51	[M + H]+	Hexadecanamide (Palmitic amide)	C_16_H_33_NO	7,806,388.659
	278.2449		[M + Na]+			1,763,628.958
	511.5185	29.52	[2M + H]+			449,241.592
	533.5006		[2M + Na]+			363,277.043
	294.2182		[M + K]+			35,062.909
22	297.2893		[M + H-NH_3_]+	Solacaproine	C_18_H_39_N_3_O	86,740.906
	314.3049		[M + H]+			52,611.286
23	285.2879	29.77	[M + H]+	Ethyl palmitate	C_18_H_36_O_2_	58,073.530
**B. Negative ionization mode**						
24	111.0088	0.75	[M-H]-	3-Methyl-2-5-furandione	C_5_H_4_O_3_	100.285
25	117.01932	0.83	[M-H]-	Succinic acid	C_4_H_6_O_4_	12,012
26	353.08783	2.21	[M-H]-	Chlorogenic acid	C_16_H_18_O_9_	258,912
27	207.050913	3.29	[M-H]-	4-O-Methylglucuronic acid	C_7_H_12_O_7_	4598.659
28	131.07127	3.33	[M-H]-	Ethyl 3-hydroxy-butanoate	C_6_H_12_O_3_	2582
29	179.03492	3.86	[M-H]-	Caffeic Acid	C_9_H_8_O_4_	5934
30	175.04000	4.23	[M-H-COCH_2_]-	4-Methylumbelliferyl acetate	C_12_H_10_O_4_	27,106.992
	217.05106		[M-H]-			4492.950
31	176.01133	6.56	[M-H-CH_3_]-	Scopoletin	C_10_H_8_O_4_	26,388.932
	191.03486		[M-H]-			43,250.386
	259.02198		[M-H + NaCOOH]-			9135.296
32	609.1457	9.19	[M-H]-	Rutin	C_27_H_30_O_16_	25,850
33	463.08799	10.39	[M-H]-	Hyperoside	C_21_H_20_O_12_	18,198
34	277.21675	29.84	[M-H]-	Linolenic acid	C_18_H_30_O_2_	51,774.794
	345.20465		[M-H + NaCOOH]-			4077.093

## Data Availability

The authors declare that the data supporting the findings of this study are available within the paper. If any raw data files are needed in another format, they are available from the corresponding author upon reasonable request.

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
