# Peer review of "Mandragora autumnalis: Phytochemical Composition, Antioxidant and Anti-Cancerous Bioactivities on Triple-Negative Breast Cancer Cells"

_ijms, 2025, doi:10.3390/ijms26178506_

Round 1
Reviewer 1 Report
Comments and Suggestions for Authors
- The focus of the current study on Mandragora autumnalis is noteworthy; however, the novelty should be more clearly emphasized by comparing the findings with previous research on the plant’s anticancer activity, if such studies exist.
- The rationale for selecting MDA-MB-231 cells as the specific model should be stated more explicitly in the abstract or the introduction section.
- The list of biomarkers (Ki67, MMP-9, STAT-3) could be presented in a clearer format to avoid ambiguity.
- The ROS and antioxidant results are described only briefly; providing more precise details would strengthen the impact. Additionally, the statement that MAE increased ROS levels while also exhibiting antioxidant properties appears contradictory and requires clarification or a brief explanation of the possible dual mechanism.
- It is also unclear whether the observed effects on p53, Ki67, MMP-9, and STAT-3 are direct or secondary to upstream molecular changes; a concise mechanistic link would enhance understanding.
- Furthermore, several typographical and punctuation errors are present throughout the manuscript.
- The phrase “radical scavenging activity” should include the actual percentage or degree observed in the DPPH assay.
- Provide the limitations of the current study.
Minor English editing is advised.
Author Response
Dear Editor, dear Reviewer,
We sincerely thank the reviewers for their time, valuable insights, and constructive feedback. Their thoughtful comments have been instrumental in improving the clarity, rigor, and overall quality of our manuscript.
Please find below our answers to each comments
Comments and Suggestions for Authors
- C1: “The focus of the current study on Mandragora autumnalis is noteworthy; however, the novelty should be more clearly emphasized by comparing the findings with previous research on the plant’s anticancer activity, if such studies exist”.
A1: Thank you for your valuable feedback. We acknowledge the importance of clearly highlighting the novelty of our work. While limited studies (only two) have addressed the anticancer activity of Mandragora autumnalis, all existing literature on the anticancer activity of Mandragora autumnalis, is incorporated in the discussion (highlighted in yellow), and a direct comparison with our findings is added.
The added section:
The novelty of the present study lies in its focus on elucidating the molecular mechanisms and chemopreventive pathways of Mandragora autumnalis ethanolic leaf extract, specifically against triple-negative breast cancer MDA-MB-231 cells, a highly aggressive and treatment-resistant subtype. While previous studies mainly assessed the broad-spectrum cytotoxicity of Mandragora autumnalis extracts from flowers, fruits, whole plants, or crude leaf preparations against a variety of cancer cell lines, including hormone-receptor-positive breast cancer (MCF-7), lung cancer (A549), colon cancer (HCT-116), and murine mammary sarcoma (EMT6/p), these earlier studies primarily concentrated on general antiproliferative activity and reported results like decreased VEGF expression, low cytotoxicity toward normal cells, and tumor size reduction in animal models. The present study, on the other hand, advances the field by combining phytochemical profiling with in-depth mechanistic analyses. It reveals that MAE modulates important cancer hallmarks, such as p53 activation, suppression of proliferation factor Ki67, inhibition of invasion and metastasis mediators MMP-9 and STAT-3, and alteration of cell cycle dynamics, in addition to exerting concentration and time-dependent cytotoxic effects. Our finding emphasizes that antioxidant pre-treatment using N-acetyl cysteine (NAC) could prevent ROS induced by MAE, highlight a ROS-mediated apoptosis pathway, and provide a level of mechanistic detail not found in previous studies. This study contributes to a better understanding of Mandragora autumnalis's anticancer potential by advancing from descriptive cytotoxicity to molecularly targeted insights that are pertinent to the creation of novel TNBC treatments.
- C2: “The rationale for selecting MDA-MB-231 cells as the specific model should be stated more explicitly in the abstract or the introduction section.”
A2: We appreciate this valuable suggestion. In response, we revised the introduction to clearly articulate the rationale for selecting MDA-MB-231 cells, underscoring their significance as a well-established triple-negative breast cancer model.
The section added to the introduction:
The MDA-MB-231 cell line was chosen for this study due to its high clinical relevance as a representative model of TNBC. TNBC constitutes 15–20% of all breast cancer cases and is associated with rapid progression, early metastasis, poor prognosis, and the absence of effective targeted therapies. MDA-MB-231 cells exhibit a basal-like molecular profile and harbor TP53 mutations, closely reflecting the genetic landscape of TNBC tumors in patients. Functionally, they possess a highly invasive and metastatic phenotype, characterized by mesenchymal morphology, elevated expression of pro-metastatic mediators such as MMP-9, and activation of oncogenic signaling pathways, including STAT3. Additionally, their well-documented chemoresistance to conventional therapies mirrors the therapeutic challenges faced in clinical TNBC management, making them a stringent and predictive platform for evaluating novel anticancer agents. These combined genetic, phenotypic, and clinical parallels underscore the suitability of MDA-MB-231 cells as an optimal in vitro model for mechanistic investigations and preclinical testing of potential TNBC-targeted treatments.
- C3: “The list of biomarkers (Ki67, MMP-9, STAT-3) could be presented in a clearer format to avoid ambiguity”.
A3: Thank you for pointing this out. The manuscript is revised to present the list of biomarkers, including Ki67, MMP-9, and STAT-3, in a clearer format.
The revised biomarkers in the abstract:
This extract caused a significant decrease in the expression of Ki-67 (a cellular proliferation marker), MMP-9 (matrix metalloproteinase-9, an enzyme involved in extracellular matrix degradation and metastasis), and STAT-3 (a transcription factor regulating cell growth and survival).
- C4: “The ROS and antioxidant results are described only briefly; providing more precise details would strengthen the impact. Additionally, the statement that MAE increased ROS levels while also exhibiting antioxidant properties appears contradictory and requires clarification or a brief explanation of the possible dual mechanism.”
A4: Thank you for this valuable feedback. We expanded the description of the ROS and antioxidant results to include more precise methodological and quantitative details, thereby enhancing the clarity and impact of this section. Additionally, we revised the discussion and addressed the apparent contradiction between increased ROS levels and antioxidant activity, providing a brief explanation of the possible dual mechanism.
This is the possible dual mechanism, which was added to the discussion section:
The main intracellular antioxidant, glutathione, is directly derived from N-acetylcysteine (NAC). NAC is commonly used as a mechanistic probe in cancer research because it can reduce ROS accumulation and restore cell viability when a compound causes cancer cell death by generating reactive oxygen species (ROS). By using this method, our findings ascertained that oxidative stress plays a significant role in mediating the MAE's cytotoxic effects. In that context, MAE simultaneously causes ROS production in MDA-MB-231 cells while exhibiting antioxidant activity in cell-free radical scavenging assays. The context- and dose-dependent redox characteristics of many phytochemicals, which can function as antioxidants in healthy physiological settings but have pro-oxidant, cytotoxic effects on cancer cells that are already under a lot of oxidative stress, are reflected in this dual behavior.
- C5: “It is also unclear whether the observed effects on p53, Ki67, MMP-9, and STAT-3 are direct or secondary to upstream molecular changes; a concise mechanistic link would enhance understanding”.
A5: Thank you for highlighting this important point. The discussion was revised to clarify whether the observed effects on p53, Ki67, MMP-9, and STAT-3 are likely direct or secondary to upstream molecular events. Where possible, we will provide a concise mechanistic link based on our findings and relevant literature, thereby enhancing the reader’s understanding of the potential signaling pathways involved.
This is the revised section:
The effects of MAE on p53, Ki-67, MMP-9, and STAT-3 are likely secondary to upstream molecular changes, particularly oxidative stress. The extract induces reactive oxygen species (ROS) production in MDA-MB-231 cells, and this effect is mitigated by the antioxidant N-acetylcysteine (NAC), suggesting that ROS generation is a critical mediator of the observed cellular responses. The induction of p53 phosphorylation following MAE treatment may represent a cellular response to oxidative DNA damage, leading to cell cycle arrest and apoptosis. Concurrently, the downregulation of Ki-67, MMP-9, and STAT-3 indicates a suppression of cell proliferation, invasion, and survival pathways, which are commonly regulated by p53 and other stress-responsive signaling mechanisms. Therefore, the modulation of these markers appears to be a consequence of ROS-induced signaling cascades rather than direct interactions with the extract components.
- C6: “Furthermore, several typographical and punctuation errors are present throughout the manuscript”.
A6: Thank you for bringing this to our attention. We carefully proofread the manuscript and corrected all typographical and punctuation errors to ensure clarity, accuracy, and consistency throughout the text.
- C7: “The phrase “radical scavenging activity” should include the actual percentage or degree observed in the DPPH assay”.
A7: Thank you for your suggestion. We revised the manuscript to include the actual percentage observed in the DPPH assay when describing the radical scavenging activity in the abstract, providing precise quantitative data to enhance clarity and accuracy.
- C8: “Provide the limitations of the current study”.
A8: Thank you for this important suggestion. A dedicated section outlining the limitations of the current study, to provide a balanced perspective on the findings was added.
The limitations section:
Although, the study's in vitro results show that MAE has anticancer potential against MDA-MB-231 cells, it lacks in vivo validation, which is necessary to evaluate the extract's safety, pharmacokinetics, and efficacy in a setting that is more physiologically relevant. Mechanistic insight and reproducibility were also limited because, despite the extract's phytochemical content and antioxidant qualities being described, no fractionation or identification of the precise bioactive compounds causing the observed effects was carried out. Also, the extract's safety, effectiveness, and dosage in humans should be tested in the form of clinical trials. Moreover, a lack of precise quantitative analysis of the individual compounds in Mandragora autumnalis extract. While the TPC, TFC, and LC-MS data provide information on relative concentrations, they do not allow the determination of the exact amounts of bioactive compounds reaching the cellular level or whether these concentrations are physiologically achievable for therapeutic purposes. We acknowledge this important point and agree that precise quantification would provide clearer insight into the extract’s physiological relevance. Hence, to completely validate Mandragora autumnalis's anticancer potential, these limitations highlight the necessity of in vivo research, the isolation of bioactive compounds, quantitative analyses, and clinical evaluation.
C9: “Comments on the Quality of the English Language
Minor English editing is advised”.
A9: Thank you for your feedback. The manuscript was carefully reviewed and underwent minor English language editing to improve clarity, grammar, and overall readability.
Regards
Dr M Maresca
Reviewer 2 Report
Comments and Suggestions for Authors
Review IJMS
This manuscript aims to demonstrate the anticancer activity of MAE in vitro in breast cancer cells, evaluating the various mechanisms involved. It is a data-rich study, very well structured in its methods and design. However, there are also some uncertainties that need clarification and some modifications that need to be made to make it suitable for publication in IJMS.
Specific comments:
Abstract
Line 28 – 32 “While MAE exhibited radical scavenging activity in the DPPH assay, it also caused an increase in reactive oxygen species (ROS) production in MDA-MB-231 cells, however, the antioxidant N- acetyl cysteine (NAC) blocked this effect.”. Actually, no experiment has been assessed to demonstrate that MAE induces the production of ROS in cells, but only indirect data from the inhibition of MAE toxicity by NAC, which is an antioxidant but could as well act through mechanisms other than scavenging and inhibition of oxidative stress. Please rephrase this sentence.
Introduction
Lines 90-96 This section, which defines the scope of the work, should be improved because it appears partially disconnected from what is stated in the introduction part.
Results
- Lines 145 – 147 “The TPC, TFC, and LC-MS data are consistent with these results, indicating that the extract's antioxidant capacity increases with phenolic and flavonoid concentration.” Are there any statistical data that confirm this trend, or is it just a hypothesis? Please explain and implement this section.
- The first section shows how the extract is capable of exhibiting antioxidant activity, while the subsequent section establishes that the extract is capable of exerting an oxidant action at the cellular level through an indirect method, namely the reversal of the extract's toxicity by NAC. How do these two apparently contradictory findings correlate? A test like the DFCH-DA one would be useful to demonstrate that the extract does indeed induce ROS production at the cellular level, perhaps due to the fact that at high concentrations, the polyphenols contained in it can act as pro-oxidants rather than antioxidants as reported elsewhere.
- The abbreviation MDA-MB-231 cells should always be reported in the text and not shortened to “MDA cells”.
- A precise quantitative analysis, not just of “intensity”, of the compounds present in the extract would be necessary to understand what concentrations they reach at the cellular level and whether these concentrations are physiologically achievable or too high for potential therapeutic use.
Discussion
- In my opinion, the Discussion is too long and should be condensed by focusing only on the most important aspects that emerged from this scientific work. Furthermore, it could, if possible, be supplemented with a short section on the limitations and strengths of this research.
- Table 4 would be helpful if presented as an explanatory image, which could be more impactful for the reader.
Author Response
Dear Editor, dear Reviewer,
We sincerely thank the reviewers for their time, valuable insights, and constructive feedback. Their thoughtful comments have been instrumental in improving the clarity, rigor, and overall quality of our manuscript.
Please find below our answers to each comments:
Comments and Suggestions for Authors
C1: “This manuscript aims to demonstrate the anticancer activity of MAE in vitro in breast cancer cells, evaluating the various mechanisms involved. It is a data-rich study, very well structured in its methods and design. However, there are also some uncertainties that need clarification and some modifications that need to be made to make it suitable for publication in IJMS”.
A1: We would like to thank and sincerely appreciate the reviewer’s thorough evaluation and constructive feedback. In response, we addressed the identified uncertainties and implemented the suggested modifications to clarify key points and further strengthen the manuscript, ensuring it meets the expected standards of rigor and clarity.
C2: “Line 28 – 32 “While MAE exhibited radical scavenging activity in the DPPH assay, it also caused an increase in reactive oxygen species (ROS) production in MDA-MB-231 cells, however, the antioxidant N- acetyl cysteine (NAC) blocked this effect.”. Actually, no experiment has been assessed to demonstrate that MAE induces the production of ROS in cells, but only indirect data from the inhibition of MAE toxicity by NAC, which is an antioxidant but could as well act through mechanisms other than scavenging and inhibition of oxidative stress. Please rephrase this sentence”.
A2: Thank you for your comment.
This is the rephrased sentence:
Although MAE exhibited 55% radical scavenging activity at higher concentrations in the DPPH assay, the attenuation of its cytotoxic effects in MDA-MB-231 cells with N-acetylcysteine (NAC) co-treatment suggests a potential role of oxidative stress.
C3: “Introduction
Lines 90-96 This section, which defines the scope of the work, should be improved because it appears partially disconnected from what is stated in the introduction part.”
A3: Thank you for this observation. Lines 90–96 were revised to better define the scope of the work and ensure it is aligned with the statements and context provided in the introduction, thereby improving coherence and flow.
This is the revised section:
Mandragora species has a long history of use as a traditional herbal remedy, as evidenced by the use of its roots, fruits, and leaves to treat conditions like ulcers, inflammation, insomnia, and eye disorders. The plant's biological activities and phytochemical composition, especially its anticancer potential, have not been thoroughly studied despite its historical significance. In order to address this gap, the current study characterized the phytochemical profile of Mandragora autumnalis ethanolic leaf extract (MAE) using qualitative assays and Liquid Chromatography-Mass Spectrometry (LC-MS). MAE's anticancer activity against the MDA-MB-231 cell line was assessed, building on the plant's established bioactive qualities and traditional therapeutic relevance. Also, the effect of MAE on cell proliferation, apoptosis induction, and several molecular markers of metastasis, such as adhesion, invasion, cell cycle regulation, angiogenesis, migration, and aggregation, was tested. The historical medicinal use of the plant, its phytochemical components, and its possible mechanisms for preventing the progression of TNBC are all directly linked by this integrative approach.
C4: “Results
- Lines 145 – 147 “The TPC, TFC, and LC-MS data are consistent with these results, indicating that the extract's antioxidant capacity increases with phenolic and flavonoid concentration.” Are there any statistical data that confirm this trend, or is it just a hypothesis? Please explain and implement this section”.
A4: Thank you for this valuable comment. Qualitative analyses of the MAE revealed substantial levels of phenolic and flavonoid compounds, as confirmed by TPC, TFC, and LC-MS data. These findings suggest that higher concentrations of these phytochemicals may be linked to enhanced antioxidant activity in the DPPH assay. However, no formal statistical correlation was performed, and therefore, this relationship is interpreted as a qualitative trend rather than a confirmed quantitative association. This section was implemented in the text.
C5: -“The first section shows how the extract is capable of exhibiting antioxidant activity, while the subsequent section establishes that the extract is capable of exerting an oxidant action at the cellular level through an indirect method, namely the reversal of the extract's toxicity by NAC. How do these two apparently contradictory findings correlate? A test like the DFCH-DA one would be useful to demonstrate that the extract does indeed induce ROS production at the cellular level, perhaps due to the fact that at high concentrations, the polyphenols contained in it can act as pro-oxidants rather than antioxidants as reported elsewhere”.
A5: Thank you for this insightful observation. We acknowledge the apparent contradiction between the extract’s antioxidant activity and its inferred pro-oxidant effect at the cellular level. The discussion was revised to clarify this point, noting that the current evidence for ROS induction is indirect (via NAC-mediated protection) and that direct assessment using assays such as DCFH-DA would be valuable in future studies (included in the text). We also discussed the possibility that, at higher concentrations, polyphenols in the extract may act as pro-oxidants, as reported in previous literature, providing a plausible explanation for these dual effects.
The revised section in the discussion:
The main intracellular antioxidant, glutathione (GSH), is directly derived from N-acetylcysteine (NAC). NAC is commonly used as a mechanistic probe in cancer research because it can reduce ROS accumulation and restore cell viability when a compound causes cancer cell death by generating reactive oxygen species (ROS). By using this method, our findings ascertained that oxidative stress plays a significant role in mediating the MAE's cytotoxic effects. In that context, MAE simultaneously causes ROS production in MDA-MB-231 cells while exhibiting antioxidant activity in cell-free radical scavenging assays. The context- and dose-dependent redox characteristics of many phytochemicals, which can function as antioxidants in healthy physiological settings but have pro-oxidant, cytotoxic effects on cancer cells that are already under a lot of oxidative stress, are reflected in this dual behavior. On the other hand, a test like the DFCH-DA would be useful to demonstrate that the extract does induce ROS production at the cellular level
C6: “The abbreviation MDA-MB-231 cells should always be reported in the text and not shortened to “MDA cells””.
A6: Thank you for your valuable comment. The cells were reported as MDA-MB-231 cells in the entire manuscript.
C7: “A precise quantitative analysis, not just of “intensity”, of the compounds present in the extract would be necessary to understand what concentrations they reach at the cellular level and whether these concentrations are physiologically achievable or too high for potential therapeutic use”.
A7: Thank you for this important suggestion. We fully agree that precise quantitative analysis of the individual compounds would provide clearer insight into their physiological relevance. However, due to current limitations in access to analytical standards and equipment, this study was limited to qualitative profiling. Nonetheless, the extract was tested at concentrations of 100 and 200 µg/mL, which are commonly used in in vitro assays and fall within a range that is considered biologically relevant in similar studies. These concentrations were chosen to reflect potential therapeutic levels without exceeding cytotoxic thresholds. Future studies should include detailed quantification and pharmacokinetic modeling to support translational potential, which is acknowledged in the limitations section.
C8: “Discussion
- In my opinion, the Discussion is too long and should be condensed by focusing only on the most important aspects that emerged from this scientific work. Furthermore, it could, if possible, be supplemented with a short section on the limitations and strengths of this research”.
A8: Thank you for your thoughtful feedback. The discussion was condensed as much as possible to focus on the most significant findings of the study, ensuring clarity and conciseness. Additionally, a brief section outlining the key strengths and limitations of the research was added to provide a balanced perspective on the study’s impact and context.
C9: “Table 4 would be helpful if presented as an explanatory image, which could be more impactful for the reader.”
A9: Thank you for this constructive suggestion. We converted Table 4 into an explanatory scheme (Figure 11) to enhance visual clarity and improve the reader’s comprehension of the data.
Regards
Dr M Maresca
Round 2
Reviewer 1 Report
Comments and Suggestions for Authors
The revised version is now much improved, and responses are more satisfactory. Hence, it is recommended for publication.
Author Response
Dear Reviewer,
Many thanks for your comments that helped us to improve the manuscript
regards
Dr M Maresca
Reviewer 2 Report
Comments and Suggestions for Authors
The authors have satisfactorily addressed all of my concerns.
Author Response

(The authors gave the same response as above.)
